# Theoretical Analysis of Sparse Optimization with Reparameterization, Weight Decay, and Adaptive Learning Rate

**Huangyu Xu** [1 2]  **Jingqin Yang** [3]  **Qianqian Xu** [1 4]  **Jiaye Teng** [5 6 *]

## Abstract

Sparse optimization is a fundamental challenge in various practical applications. A popular approach to sparse optimization is $\ell_p$ regularization. However, it may encounter optimization instability due to the unbounded gradients when $0 < p < 1$. In this paper, we introduce a novel approach to sparse optimization termed ReWA, based on *R*eparameterization, *W*eight decay, and *A*daptive learning rate. ReWA is closely connected to $\ell_p$-regularization, yet it unveils a distinct optimization landscape that helps mitigate instability issues. Experiments on CIFAR-10 and ImageNet with ResNets demonstrate that ReWA leads to significant sparsity improvements over the $\ell_1$-regularization approach while preserving test accuracy.

## 1. Introduction

This paper focuses on sparse training, which involves identifying solutions with a minimal number of non-zero coefficients while adhering to specified constraints (Natarajan, 1995b; Candes & Tao, 2006; Lustig et al., 2007; Zhang et al., 2015). The most direct approach to sparse training relies on $\ell_0$ regularization. However, directly solving $\ell_0$ regularization poses significant challenges due to its inherent non-continuity (Natarajan, 1995a; Chen et al., 2017; Mousavi et al., 2019). A practical approach is to relax $\ell_0$

[1]State Key Laboratory of AI Safety, Institute of Computing Technology, Chinese Academy of Sciences, Beijing, China [2]School of Computer Science and Technology, University of Chinese Academy of Sciences, Beijing, China [3]IIIS, Tsinghua University, Beijing, China [4]Beijing Academy of Artificial Intelligence (BAAI), Beijing, China [5]School of Statistics and Management, Shanghai University of Finance and Economics, Shanghai, China [6]Institute of Data Science and Statistics, Shanghai University of Finance and Economics, Shanghai, China. Correspondence to: Jiaye Teng <tengjiaye@sufe.edu.cn>.

*Proceedings of the 43$^{rd}$ International Conference on Machine Learning*, Seoul, South Korea. PMLR 306, 2026. Copyright 2026 by the author(s).

regularization with $\ell_1$ regularization:

$$\min_{\boldsymbol{x}} L_1(\boldsymbol{x}) = f(\boldsymbol{x}) + \lambda_1 \|\boldsymbol{x}\|_1, \qquad ([B1])$$

where $\boldsymbol{x} \in \mathbb{R}^d$ denotes a variable, $f(\cdot) \in \mathbb{R}$ denotes an objective function, and $\lambda_1$ denotes a regularization parameter. Extensive research in compressed sensing and LASSO has both theoretically and empirically showcased the effectiveness of the $\ell_1$ regularization (Tibshirani, 1996; Zou, 2006; Candes & Tao, 2006; Candes, 2008). Nonetheless, a line of works argues that $\ell_1$ regularization may introduce bias, potentially limiting its applicability in broader scenarios (Zhang & Huang, 2008; Van de Geer et al., 2014; Chartrand, 2007).

To overcome the limitations of $\ell_1$ regularization, researchers have explored diverse approaches. These include $\ell_p$ regularization with $0 < p < 1$ (Frank & Friedman, 1993; Fu, 1998), combinational techniques (Portilla & Mancera, 2007; Feng et al., 2013), implicit bias towards sparsity (Gunasekar et al., 2018; Arora et al., 2019), and smooth approximations for the challenging $\ell_0$ regularization (Louizos et al., 2018). Among these, the $\ell_p$ regularization branch stands out since (a.) it offers the flexibility to be deployed into general loss functions without mandating explicit problem definitions, and (b.) it comes with accompanying theoretical guarantees (Chartrand, 2007; Chartrand & Staneva, 2008; Saab et al., 2008; Chartrand, 2009; Zheng et al., 2017). We note that other nonconvex methods such as SCAD (Fan & Li, 2001), MCP, and adaptive Lasso also address the bias of $\ell_1$; we view $\ell_p$ as a complementary rather than competing approach, and discuss the relationship in detail in Appendix B. The $\ell_p$ regularization approach is formally defined as follows, where $\lambda_2$ denotes a regularization parameter.

$$\min_{\boldsymbol{x}} L_p(\boldsymbol{x}) = f(\boldsymbol{x}) + \lambda_2 \|\boldsymbol{x}\|_p^p. \qquad ([Bp])$$

However, optimizing $\ell_p$ regularization remains difficult due to its unbounded gradient and non-smooth nature, limiting its practicality primarily to linear regimes (Marjanovic & Solo, 2012; 2013; Wen et al., 2018). Recent works (Hoff, 2017; Kolb et al., 2023) have attempted to reparameterize $\ell_p$ regularization with [Cp]. For some given odd $K > 0$,

consider the form

$$\min_{\boldsymbol{y}_1, \cdots, \boldsymbol{y}_K} L'_p(\boldsymbol{y}_1, \cdots, \boldsymbol{y}_K)$$
$$= f(\boldsymbol{y}_1 \odot \boldsymbol{y}_2 \odot \cdots \odot \boldsymbol{y}_K) + \frac{\lambda}{2} \sum_{i \in [K]} \|\boldsymbol{y}_i\|_2^2, \quad \text{([Cp])}$$

where $\odot$ denotes the element-wise product (*i.e.*, Hadamard product), $\boldsymbol{y}$ denotes the reparameterized vector (the subscript is omitted when the context is clear), and $\lambda$ denotes the regularization parameter. The local and global minimizers of [Cp] closely relate to those of $\ell_p$ regularization with $p = 2/K$ given a proper $\lambda$ (Hoff, 2017; Kolb et al., 2023). Notably, [Cp] has bounded gradients around zero, making the optimization easier compared to $\ell_p$ regularization. Nonetheless, this approach may encounter optimization instability in general scenarios, posing challenges for direct application in complex real-world datasets (See Figure 1, Example 3.2 and Theorem 3.10 for more details).

In this paper, we propose a new sparse optimization method called ReWA[1], as illustrated in Algorithm 1. ReWA comprises three components: *R*eparameterization, *W*eight decay and *A*daptive learning rate. Different from the conventional techniques which directly operate on $\boldsymbol{x} \in \mathbb{R}^d$, ReWA projects the vector $\boldsymbol{x}$ to the power of $1/K$ element-wisely inspired by loss form [Cp], yielding $\boldsymbol{y} \in \mathbb{R}^d$. It then iterates on the vector $\boldsymbol{y}$, using weight decay and adaptive learning rate. The iteration of ReWA is derived based on the reparameterized loss form [Cp], which is closely related to $\ell_p$ regularization where $p \in (0, 1)$. Hence, ReWA is well-positioned for superior performance, given the tendency for $\ell_p$ approximation to outperform $\ell_1$ approximation.

Notably, ReWA is distinct from directly applying GD/SGD on loss form [Cp]. The key distinction comes from the **coordinate-wise adaptive learning rate**. This adaptive learning rate ensures stable training and enables its application to complex real-world datasets (We refer to Example 3.2 for further details). However, a question arises: does the new iteration with the adaptive learning rate still maintain a connection to the $\ell_p$ regularization with $0 < p < 1$? We prove in Theorem 3.3 that ReWA relates to the following regularizer. This creates an *empirical tip* on properly setting $\epsilon$ and $M$ which leads to an $\ell_p$ regularization with $0 < p < 1$.

$$R(\boldsymbol{x}) = \frac{K}{1 - M + K} \|\boldsymbol{x}\|_{1+(1-M)/K}^{1+(1-M)/K} + \epsilon \frac{K}{2 - M} \|\boldsymbol{x}\|_{(2-M)/K}^{(2-M)/K},$$

We further provide additional theoretical evidence on each component of ReWA, including reparameterization (Theorem 3.7), weight decay (Theorem 3.9), and adaptive learning rate (Theorem 3.10). Besides, Theorem 3.14 further presents advantages of $\ell_p$ regularizers. Empirical verifications on both synthetic (Section 4.1) and real-world datasets

---

[1]Code: github.com/childofcuriosity/rewa.

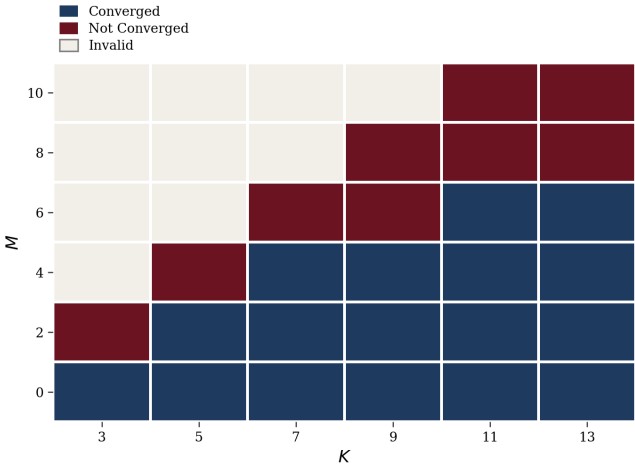

*Figure 1.* The ablation study of $K$ and $M$ on linear regression models. Blue means the returned test loss could be small, red means the test loss is large, and white means that $M > K - 1$.

(CIFAR-10 and ImageNet in Section 4.2) demonstrate that ReWA consistently returns sparse solutions.

*Role of sparse optimization.* One of the classic applications revolves around sparse signal recovery, notably prevalent in the medical domain (Lustig et al., 2007; 2008; Marvasti et al., 2012; Stanković et al., 2019). In the realm of modern machine learning, sparse optimization is crucial in enhancing the effectiveness of network pruning strategies, contributing to model efficiency and performance improvement (Han et al., 2015; 2016; Louizos et al., 2018; Savarese et al., 2020; Xia et al., 2023). Recent studies also highlight the potential of sparsity in various facets: augmenting fine-tuning performance in large language models (Panigrahi et al., 2023), overcoming catastrophic forgetting issues in continual learning scenarios (Kang et al., 2022; Schwarz et al., 2021), and bolstering out-of-distribution performance capabilities (Zhang et al., 2021).

## 2. Related Work

**Reparameterization-based sparsification.** Reparameterizing a variable as a product of auxiliary variables, combined with weight decay, induces sparse regularization (Grandvalet, 1998; Hoff, 2017; Kolb et al., 2023). Even without weight decay, reparameterization exhibits an implicit bias toward sparsity in specific settings such as linear models and matrix factorization (Gunasekar et al., 2018; Arora et al., 2019; Woodworth et al., 2020). Recent works further develop practical algorithms (Jacobs & Burkholz, 2025; Gadhikar et al., 2026) and study the dynamics when explicit regularization is added (Jacobs et al., 2025).

$\ell_p$ **regularization theory.** $\ell_p$ regularization ($0 < p < 1$) provides stronger sparsity guarantees than $\ell_1$ and better approximates $\ell_0$ as $p \to 0$ (Chartrand, 2007; Chartrand

---

**Algorithm 1** ReWA algorithm

---

**Require:** object function $f(\boldsymbol{x})$ with gradient $\nabla f(\boldsymbol{x})$, initialization $\boldsymbol{x}(0)$, weight decay parameter $\lambda$, learning rate scheduler $\eta_t \in \mathbb{R}$, element-wise power function $(\cdot)^K$, parameter $M, \epsilon \in \mathbb{R}$

1: Set $\boldsymbol{y}(0) = \boldsymbol{x}(0)^{1/K}$
2: **for** $t \in [T]$ **do**
3:     $\boldsymbol{y}(t+1) = (1 - \lambda\eta_t)\boldsymbol{y}(t) - \eta_t \frac{\boldsymbol{y}^M(t)}{\boldsymbol{y}^{K-1}(t)+\epsilon\mathbf{1}} \odot \boldsymbol{y}^{K-1}(t) \odot \nabla f(\boldsymbol{y}^K(t))$
4:     (Optional) $\boldsymbol{x}(t+1) = \boldsymbol{y}^K(t+1)$
5: **end for**
**Ensure:** $\boldsymbol{x}(T) = \boldsymbol{y}^K(T)$ .

---

& Staneva, 2008). However, its non-smooth, unbounded-gradient nature makes direct optimization challenging beyond linear regimes (Marjanovic & Solo, 2012; Fan & Li, 2001).

**Adaptive learning rate methods.** Adaptive optimizers such as Adam (Kingma, 2014) and AdamW (Loshchilov & Hutter, 2019) have become standard in deep learning by scaling each coordinate's update by the square root of accumulated squared gradients. This per-coordinate normalization naturally counteracts the vanishing-gradient effect introduced by high-order reparameterizations, motivating the adaptive learning rate design in ReWA.

We provide detailed discussions and additional related works in Appendix A.

## 3. ReWA

This section introduces the fundamental concepts of ReWA proposed in Algorithm 1. We first introduce the derivation of ReWA in Section 3.1, including the relationships among $\ell_p$ regularizations, [Cp] and ReWA. We subsequently provide the corresponding implicit regularizer of ReWA in Section 3.2. We finally provide additional theoretical verifications on each component of ReWA in Section 3.4.

### 3.1. Insights Behind ReWA

This section introduces insights of ReWA. Firstly, Section 3.1.1 establishes the relationship between [Cp] and $\ell_p$ regularizations regarding the global minimizers, local minimizers, and stationary points. Subsequently, Section 3.1.2 demonstrates that directly applying GD / SGD on [Cp] with a constant learning rate may lead to optimization instability issues. To address this, we introduce a novel adaptive learning rate in Equation 3, based on the analysis in Example 3.2.

#### 3.1.1. FROM $\ell_p$ REGULARIZATION TO [CP]

We start from the reparameterization form [Cp] mentioned before (Hoff, 2017; Kolb et al., 2023). It has been proven that the local and global minimizers of [Bp] ($\ell_p$ regularizations) and [Cp] are closely related, given that $p = 2/K \in$

$(0, 1)$. Therefore, minimizing [Cp] can be a meaningful surrogate of minimizing [Bp]. We make a slight extension to the results for stationary points (See Lemma 3.1), which is useful under non-convex training regimes.

**Lemma 3.1** (Relationship between [Cp] and [Bp], the first two arguments are from (Hoff, 2017; Kolb et al., 2023)). *For a differentiable function $g(\boldsymbol{x})$, a stationary point is a variable $\boldsymbol{x}$ such that $\nabla g(\boldsymbol{x}) = 0$, and a substationary point is a variable with $\boldsymbol{x} \odot \nabla g(\boldsymbol{x}) = 0$. Consider problem [Cp] with parameter $\lambda$ and problem [Bp] with parameter $K\lambda$,*

- *The global minimizer of [Cp], if exists, must be the global minimizer of [Bp].*
- *The local minimizer of [Cp], if exists, must be the local minimizer of [Bp].*
- *If [Cp] is differentiable, its stationary point, if exists, must be the substationary point of [Bp].*

Lemma 3.1 sketches the relationships between [Cp] and [Bp] on the global minimizers, local minimizers, and stationary points, where [Cp] serves as a smooth surrogate sharing the same minimizers as [Bp]. Therefore, optimizing [Cp] is meaningful when [Bp] is hard to optimize. However, the two landscapes are significantly different. Specifically, the original [Bp] is non-smooth with unbounded gradients around zero, while [Cp] is smooth with bounded gradients. The key difference makes [Cp] easier to optimize.

#### 3.1.2. FROM [CP] TO REWA

To minimize [Cp], one may use an SGD iteration, namely, for all $i \in [K]$,

$$\boldsymbol{y}_i(t+1) = (1 - \lambda\eta_t)\boldsymbol{y}_i(t) - \boldsymbol{\eta}_t' \odot \frac{\odot_i \boldsymbol{y}_i(t)}{\boldsymbol{y}_i(t)} \odot \nabla f(\odot_i \boldsymbol{y}_i(t)),$$
$$(1)$$

where $\boldsymbol{\eta}_t' \in \mathbb{R}^d$ denotes the adaptive learning rate which will be discussed later, $\odot_i \boldsymbol{y}_i(t) \triangleq \boldsymbol{y}_1(t) \odot \cdots \odot \boldsymbol{y}_K(t)$, and $\frac{\odot_i \boldsymbol{y}_i(t)}{\boldsymbol{y}_i(t)} \triangleq \boldsymbol{y}_1(t) \odot \cdots \boldsymbol{y}_{i-1}(t) \odot \boldsymbol{y}_{i+1}(t) \odot \cdots \boldsymbol{y}_K(t)$.

Due to the symmetry of $\boldsymbol{y}_i$ in [Cp] and for efficiency, we tie $\boldsymbol{y}_i := \boldsymbol{y}_1$ for all $i \in [K]$, reducing cost by a factor of $K$. By

denoting $\boldsymbol{y} = \boldsymbol{y}_i$ for simplicity, this leads to

$$\boldsymbol{y}(t+1) = (1 - \lambda\eta_t)\boldsymbol{y}(t) - \boldsymbol{\eta}'_t \odot \boldsymbol{y}^{K-1}(t) \odot \nabla f(\boldsymbol{y}^K(t)), \tag{2}$$

where the power is coordinate-wise. Compared to the iteration based on [Bp], the iteration in Equation 2 has bounded gradients around zero. However, if we just set $\boldsymbol{\eta}'_t = \eta\mathbf{1}$, namely, each coordinate uses the same learning rate, its corresponding training process is unstable due to the illness of the loss landscape. Specifically, the term $\boldsymbol{y}^{K-1}(t)$ may create high-order stationary points around zero for each coordinate. Therefore, when some coordinate is mistakenly trapped in the zero point, it is hard to escape from that. The following example in Example 3.2 illustrates the phenomenon, demonstrating the optimization issue encountered in the absence of an adaptive learning rate.

**Example 3.2** (Adaptive Learning Rate). Consider a one-dimensional loss $f(\boldsymbol{x}) = (\boldsymbol{x} - 1)^2$ with reparameterization $\boldsymbol{x} \in \mathbb{R}$ with $\boldsymbol{y}^K \in \mathbb{R}$. The goal is to optimize $\boldsymbol{y}$ to find the minimizer $\boldsymbol{x} = \boldsymbol{y} = 1$. We initialize at $\boldsymbol{y}(0) = -1$ and compare two optimization schemes of $\boldsymbol{y}(t+1)$:

- Adaptive: $\boldsymbol{y}(t) - \eta\nabla f(\boldsymbol{y}^K(t))$;
- Non-adaptive: $\boldsymbol{y}(t) - \eta'\boldsymbol{y}^{K-1}(t) \odot \nabla f(\boldsymbol{y}^K(t))$

It holds that:

- For adaptive learning rate with $\eta < \frac{1}{2K}$, for any $T > 0$, $|\boldsymbol{y}(T) - 1| \le 2(1 - \frac{2\eta}{K-1})^T$;
- For non-adaptive learning rate with $\eta' < 1/4$, for any $T > 0$, $|\boldsymbol{y}(T) - 1| \ge 1$.

Example 3.2 demonstrates that an adaptive learning rate helps in escaping from the zero saddle point. When the ground truth and the initialization have different signs, it is hard to escape from the zero saddle point in Equation 2, leading to a large loss. We refer the readers to Appendix C.3 for the whole proof of Example 3.2.

Due to the above discussions on Example 3.2, one needs to use a coordinate-wise adaptive learning rate to reduce the effect of $\boldsymbol{y}^K$ and stabilize the optimization process. To this end, we propose a new adaptive learning rate $\boldsymbol{\eta}'_t = \eta_t\frac{\boldsymbol{y}^M(t)}{\boldsymbol{y}^{K-1}(t)+\epsilon\mathbf{1}}$ with $\eta_t, \epsilon \in \mathbb{R}^+$ and $0 \le M < K - 1$, leading to the ReWA iteration:

$$\boldsymbol{y}(t+1) = (1 - \lambda\eta_t)\boldsymbol{y}(t)$$
$$- \eta_t\frac{\boldsymbol{y}^M(t)}{\boldsymbol{y}^{K-1}(t) + \epsilon\mathbf{1}} \odot \boldsymbol{y}^{K-1}(t) \odot \nabla f(\boldsymbol{y}^K(t)). \tag{3}$$

The adaptive learning rate can be split into three parts. We present the intuition behind the three terms below, and provide the theoretical insights in Section 3.2.

- Term $\eta_t \in \mathbb{R}$, which represents the scale of the stepsize.

- Term $\frac{1}{\boldsymbol{y}^{K-1}(t)+\epsilon\mathbf{1}}$, which eliminate the effects of $\boldsymbol{y}^{K-1}(t)$ when $\boldsymbol{y}(t)$ is large. We will further show that this will lead to an $\ell_p$ $(0 < p < 1)$ regularization with scale $\epsilon$. Similar terms can be found in Adam-like optimizers.
- Term $\boldsymbol{y}^M(t)$ with $M \in [0, K - 1]$, which maintains an tradeoff between the sparsity (large $M$) and optimization stability (small $M$). This introduces an $\ell_p$ $(0 < p < 1)$ regularization where $p$ relates to $M$. The constraint $M \le K - 1$ ensures this ratio that remains bounded as $\boldsymbol{y}(t)$ grows.

### 3.2. Implicit Bias

This section analyzes the regularizer related to the ReWA in Theorem 3.3. Notably, different from [Cp], ReWA introduces an adaptive learning rate, which potentially alters the optimization landscape, resulting in a different regularizer compared to Lemma 3.1.

**Theorem 3.3** (Implicit Bias). *Define the regularizer $R(\boldsymbol{x})$*

$$R(\boldsymbol{x}) = \frac{K}{1 - M + K}\|\boldsymbol{x}\|_{1+(1-M)/K}^{1+(1-M)/K} + \epsilon\frac{K}{2 - M}\|\boldsymbol{x}\|_{(2-M)/K}^{(2-M)/K}, \tag{4}$$

*Assume that the parameter $\boldsymbol{x}(t)$ is bounded, and the function $f(\cdot)$ is bounded and smooth. For the ReWA algorithm with an even integer $M$, it holds that*

$$\|V(\boldsymbol{x}(T)) \odot \nabla[f(\boldsymbol{x}(T)) + R(\boldsymbol{x}(T))]\|^2 \le \frac{\log T}{\sqrt{T}},$$

*where $V(\boldsymbol{x}(t)) = \frac{\boldsymbol{x}^{1+(M-2)/(2K)}(t)}{\sqrt{\boldsymbol{x}^{1-1/K}(t)+\epsilon\mathbf{1}}}$ denotes a vector satisfying that its $i$-th coordinate $V_i(\boldsymbol{x}(T)) = 0$ if and only if $\boldsymbol{x}_i(T) = 0$.*

Theorem 3.3 implies that for a sufficiently large number of iterations T, each coordinate satisfies either $\boldsymbol{x}_i(T) \approx 0$ or $\nabla[f(\boldsymbol{x}(T)) + R(\boldsymbol{x}(T))]_i \approx 0$. Therefore, $R(\boldsymbol{x})$ can be considered as the implicit regularizer of ReWA, which contains an $\ell_p$ term with $p = 1 + (1 - M)/K \in (0, 1)$, endowing ReWA with $\ell_p$'s sparsity-inducing property while maintaining optimization stability. Besides, based on Theorem 3.3, we can derive in Proposition 3.4 the conditions under which $R(\boldsymbol{x})$ incorporates an $\ell_p$ regularizer with $0 < p < 1$.

We defer clarifications on the methodology, the even-integer restriction on $M$, and the role of assumptions to Remark C.3 in the appendix.

**Proposition 3.4.** *To achieve $\ell_p$ $(0 < p < 1)$ regularization in Theorem 3.3, it suffices to satisfy one of the following two configurations:*

- *Configuration A: $\epsilon = 0$ and $M > 1$.*
- *Configuration B: $\epsilon > 0$ and $M < 2$.*

Intuitively, Configuration A makes the optimization challenging for large $M$, while Configuration B introduces an

additional $\ell_q$ $(q > 1)$ regularization. Therefore, we recommend using Configuration A for simple datasets and models (*e.g.*, synthetic data with linear models), and Configuration B for complex problems (*e.g.*, ImageNet with neural networks).

*Remark* 3.5 (The role of $M$). It is worthwhile to compare two configurations, both of which induce the same implicit bias but use different $M$.

- Configuration C: Hyperparameter $K$ and $M$ (Ours with adaptive learning rate), $\epsilon = 0$;
- Configuration D: Hyperparameter $K' = 2/(1 + (1 - M)/K)$ and $M' = K' - 1$ (Without adaptive learning rate), $\epsilon = 0$.

Both configurations lead to the same $\ell_p$ regularizer with $p = 1 + (1 - M)/K$. The key difference lies in the coefficient of $\nabla f(\boldsymbol{x})$ in ReWA. For Configuration C, the coefficient is $\boldsymbol{x}^{M/K}(t)$, and for Configuration D, it is $\boldsymbol{x}^{(M+K-1)/(2K)}(t)$. Therefore, Configuration C has a larger $K$ given $M < K - 1$, which amplifies the update ratio $(1 - \eta c)^K$ when crossing zero, suggesting ReWA can more easily escape from stationary points near zero in a single step. This provides a key insight into the benefits of our adaptive learning rate approach by introducing a separate $M$.

## 3.3. Empirical Tips

Based on the discussions above, we provide the following tips for deploying ReWA.

**Learning rate.** Beyond the coordinate-wise adaptive learning rates previously discussed, various learning rate schedules can be applied to $\eta_t$, including constant and cosine decay. Furthermore, while Algorithm 1 utilizes SGD as the base optimizer, it is not the only option. A common alternative is AdamW (Loshchilov & Hutter, 2019) (See Algorithm 2).

**Choice of $K$.** For simplicity, we set $K$ odd. However, the reparameterization remains valid for different values of $K$. For example, for $K = 2$ (even) (Woodworth et al., 2020), one can parameterize x as $\mathbf{y}_1 \odot \mathbf{y}_1 - \mathbf{y}_2 \odot \mathbf{y}_2$. General $K$. For any arbitrary $K$, the transformation can also be expressed as $\mathbf{x} = \text{sign}(\mathbf{y}) \cdot |\mathbf{y}|^K$.

**Adaptive learning rate.** Certain optimizers, such as AdamW, inherently incorporate adaptive learning rates. These methods implicitly perform coordinate-wise adjustments similar to the mechanism in ReWA (where $\mathbf{M} = 0, \epsilon \neq 0$). In these cases, explicitly adding a separate adaptive learning rate layer may be redundant. While AdamW adapts to gradient magnitude history and implicitly induces sparsity, ReWA explicitly controls sparsity via $M$ and $K$ with a provable $\ell_p$ connection, and can be applied *on top of* a base optimizer such as AdamW.

## 3.4. Additional Theoretical Evidence

This section provides additional theoretical evidence for each component of ReWA. Section 3.4.1 validates the role of the reparameterization by proving that directly deploying $\ell_p$ regularization leads to optimization instability even with valid gradient clipping. Section 3.4.2 validates the role of the weight decay by showing that one cannot achieve sparse solutions without weight decay under some regimes. Section 3.4.3 validates the role of the adaptive learning rate by finding that the solution hardly escapes from trivial saddle points without the adaptive learning rate.

### 3.4.1. REPARAMETERIZATION

This section provides theoretical evidence for the critical role of reparameterization in sparse training. Specifically, we demonstrate its superiority over a seemingly intuitive alternative: direct $\ell_p$ regularization with gradient clipping. While gradient clipping is a common strategy to mitigate the optimization instability often associated with $\ell_p$ regularizer, our analysis reveals a fundamental and unavoidable trade-off: it either results in a poor approximation of the desired regularization or suffers from excessively large gradient norms. Our analysis begins by formally defining the two prevalent clipping techniques in Definition 3.6: constant clipping and $\ell_1$ clipping.

**Definition 3.6** (Clipping). For the $\ell_p$ regularizer $\psi_p(\boldsymbol{x}) = \|\boldsymbol{x}\|_p^p = \sum_{j \in [d]} |\boldsymbol{x}[j]|^p$, its constant clipping at threshold $u \in \mathbb{R}^+$ is formulated as $\sum_{j \in [d]} \max\{|\boldsymbol{x}[j]|^p, u^p\}$, and its $\ell_1$ clipping at threshold $u \in \mathbb{R}^+$ is formulated as $\sum_{j \in [d]} \max\{|\boldsymbol{x}[j]|^p, pu^{p-1}|\boldsymbol{x}[j]| + (1-p)u^p\}$. Figure 5 provides a visualization of these two clipping methods.

We formalize the trade-off in Theorem 3.7, which measures the effectiveness of a clipped regularizer by two key metrics:

- Gradient Bound $\mathcal{E}_1 = \max_{\boldsymbol{x}} \|\nabla_{\boldsymbol{x}} \tilde{\psi}_p(\boldsymbol{x})\|$: the maximum norm of the clipped function's gradient[2];
- Approximation Error $\mathcal{E}_2 = \max_{\boldsymbol{x}} |\psi_p(\boldsymbol{x}) - \tilde{\psi}_p(\boldsymbol{x})|$: the maximum difference between the original and clipped functions.

A low value indicates optimization stability, and a low value indicates fidelity to the original regularizer. We next prove in Theorem 3.7 the tradeoff between the gradient bound and the approximation error for both constant clipping and $\ell_1$ clipping.

**Theorem 3.7.** *For the function $\psi_p(\boldsymbol{x}) = \|\boldsymbol{x}\|_p^p$ with its clipped version $\tilde{\psi}_p(\boldsymbol{x})$, Then,*

- *For constant clipping, the events $\mathcal{E}_1 \leq \sqrt{d}$ and $\mathcal{E}_2 \leq d/(2e)$ cannot hold simultaneously.*

---

[2]When $\tilde{\psi}_p(\boldsymbol{x})$ is non-differentiable, the gradient also means subgradient.

- *For $\ell_1$ clipping, if $p < 1 - \frac{1}{cd}$ where $c \geq 1$ denotes a constant, the events $\mathcal{E}_1 \leq \sqrt{d}$ and $\mathcal{E}_2 \leq \frac{1}{ce}$ cannot hold simultaneously.*

Theorem 3.7 demonstrates the shortcomings associated with directly clipping $\ell_p$ regularization, that is, a small gradient bound and a small approximation error are mutually exclusive. The core insights stem from the nature of $\psi_p(\boldsymbol{x}) = \|\boldsymbol{x}\|_p^p$, where the gradient exhibits exponential growth around zero. Consequently, clipping methods are forced into a difficult choice: either they allow for a substantial gradient norm to preserve the original function's shape, or they heavily truncate the function to bound the gradient, resulting in significant approximation errors. This theoretical limitation provides a clear motivation for why reparameterization is a more principled approach. Full proofs and additional details are provided in Appendix C.5.

### 3.4.2. WEIGHT DECAY

This section provides a theoretical justification for using explicit weight decay with reparameterization, contrasting it with approaches like PowerPropagation (Schwarz et al., 2021) that rely solely on the implicit bias of the optimization algorithm (without explicit weight decay).

A line of theoretical research has proven that for certain problems, gradient-based optimization on reparameterized models has an implicit bias towards sparse solutions, even without explicit regularization (Gunasekar et al., 2018; Arora et al., 2019). However, these analyses are often restricted to specific settings, such as matrix factorization or linear models, and crucially depend on a small initialization. Subsequent work by Woodworth et al. (2020) revealed that this sparsity-inducing bias vanishes in the large initialization regime; instead, the dynamics are governed by an implicit $\ell_2$ regularization, which does not promote sparsity.

Our analysis addresses this gap, demonstrating that explicit weight decay is essential for achieving sparsity in the general and practical setting of arbitrary initializations. We begin with a simple yet insightful example in Example 3.8 with a quadratic objective, a setting closely related to over-parameterized linear regression.

**Example 3.8.** Consider the objective function $f(\boldsymbol{x}) = \boldsymbol{x}^\top \Lambda \boldsymbol{x}$, where $\boldsymbol{x} \in \mathbb{R}^d$ and $\Lambda \in \mathbb{R}^{d \times d}$ denotes a positive semi-definite diagonal matrix with rank $R < d$. The unique sparsest global minimizer is the origin $\boldsymbol{x} = \boldsymbol{0}$. We reparameterize the input as $\boldsymbol{x} = \odot_{k \in [K]} \boldsymbol{y}_k$ and compare two loss functions with/without weight decay:

- With: $L(\boldsymbol{y}) = f(\odot_{k \in [K]} \boldsymbol{y}_k) + \lambda \sum_{k=1}^K \|\boldsymbol{y}_k\|_2^2$;
- Without: $\tilde{L}(\boldsymbol{y}) = f(\odot_{k \in [K]} \boldsymbol{y}_k)$.

Assume gradient descent with a sufficiently small learning rate converges to a stationary point. We consider a non-

zero initialization $\alpha\mathbf{1} = (\alpha, \cdots, \alpha)$ where $\alpha = \Theta(1)$. Let $\boldsymbol{y}_k(T), \tilde{\boldsymbol{y}}_k(T)$ denote the parameter after $T$ epochs, and $\boldsymbol{y}_k(\infty), \tilde{\boldsymbol{y}}_k(\infty)$ denote the parameter at convergence, respectively. Then:

- With weight decay, for a sufficiently small but non-zero penalty $\lambda$, it holds that $\|\lim_{\lambda \to 0} \odot_{k \in [K]} \boldsymbol{y}_k(\infty)\|_0 = 0$.
- Without weight decay, for any number $T > 0$, it holds that: $\|\odot_{k \in [K]} \tilde{\boldsymbol{y}}_k(T)\|_0 \geq d - R$. Furthermore, these non-sparse coordinates remain non-zero with value $\Theta(1)$.

Example 3.8 highlights that weight decay is crucial for inducing sparsity. The intuition is straightforward: the rank-deficient matrix $\Lambda$ creates a degenerated optimization landscape, leading to zero gradient along $d - R$ dimensions (null space of $\lambda$). Consequently, gradient descent on the objective $\tilde{L}(\boldsymbol{y})$ cannot update the parameters along these untrainable directions, leaving them at their large initial values. Weight decay, by contrast, applies a uniform penalty that pulls all parameters toward the origin, effectively pruning the untrainable dimensions and ensuring a sparse solution. We next generalize this insight beyond the quadratic case. The following Theorem 3.9 abstracts the core conditions under which weight decay is necessary for sparsity.

**Theorem 3.9.** *Consider an objective function $f : \mathbb{R}^d \to \mathbb{R}$ with a reparameterized input $\boldsymbol{x} = \odot_{k \in [K]} \boldsymbol{y}_k$ where each $\boldsymbol{y}_k \in \mathbb{R}^d$. Let $\boldsymbol{y}_k(T), \tilde{\boldsymbol{y}}_k(T)$ denote the parameter after $T$ epochs, and $\boldsymbol{y}_k(\infty), \tilde{\boldsymbol{y}}_k(\infty)$ denote the parameter at convergence, respectively. Assume the objective and the optimization process satisfy the following conditions:*

- *Sparse solution: The global minimizer of $f$ is at the origin[3] $\boldsymbol{x} = \boldsymbol{0}$.*
- *Low-Dimensional update subspace: For any iteration $t$, the paramter update $\mathbf{y}_k(t+1) - \mathbf{y}_k(t)$, is confined to a fixed subspace of $\mathbb{R}^d$ with dimension $R < d$.*
- *Large initialization: The optimization starts with a large, non-zero initialization for all coordinates $\mathbf{y}_k(0) = \Theta(1)$.*

*Then, for gradient descent with a sufficiently small learning rate that converges to a stationary point, it holds that:*

- *With weight decay: $f(\odot \boldsymbol{y}_k) + \lambda \sum \|\boldsymbol{y}_k\|_2^2$, it holds that: $\|\lim_{\lambda \to 0} \odot_{k \in [K]} \boldsymbol{y}_k(\infty)\|_0 = 0$ under the condition that $\lambda \geq 1/(2K)(-\sigma_{min}\nabla^2 f(\boldsymbol{x}))$ where $\sigma_{min}$ denotes the minimal eigenvalue.*
- *Without weight decay: $f(\odot \boldsymbol{y}_k)$, for any $T > 0$, it holds that: $\|\odot_{k \in [K]} \tilde{\boldsymbol{y}}_k(T)\|_0 \geq d - R$.*

Theorem 3.9 extends the insights in Example 3.8. The critical element in overparameterized models is often the existence of directions in the parameter space along which

---

[3]We assume the zero global minimizer for simplified discussions; otherwise, there is a tradeoff between sparsity and model performance.

the primary objective's gradient provides no learning signal (the *Low-Dimensional Update Subspace* assumption). Without an explicit regularizer, parameters initialized in these directions remain unchanged. Explicit weight decay solves this problem by ensuring that all parameters are driven toward zero unless counteracted by the objective function's gradient. This demonstrates its essential role in promoting sparsity in the large initialization regime. The complete proof is provided in Appendix C.6 and C.7, along with a discussion of the mildness of the Low-Dimensional Update Subspace condition (Remark C.4).

### 3.4.3. ADAPTIVE LEARNING RATE

This section illustrates the crucial role of adaptive learning rates when employing high-order reparameterizations like $x = y^K$. The chain rule for the gradient calculation, $\nabla_y f(y^K) = \nabla_x f(x) \odot K y^{K-1}$, introduces a multiplicative term $y^{K-1}$. This multiplicative term $K y^{K-1}$ shrinks dramatically when the parameter $y_i$ is near zero, causing the corresponding gradient $\nabla_y f(y^K)$ to shrink dramatically, even if the underlying gradient $\nabla_x f(x)$ is large. This effect creates a sharp and problematic saddle point at the origin, which potentially traps standard gradient-based optimizers, especially if the optimal parameter value and its initialization have opposite signs.

Adaptive methods like Adam or RMSprop mitigate this problem by normalizing the gradient update. They typically divide the update by a running average of the magnitude of past gradients. This normalization counteracts the vanishing effect of the $y^{K-1}$ term, allowing the optimizer to maintain momentum and effectively escape the saddle point at the origin. We demonstrate this mechanism with a simple one-dimensional example in Example 3.2, omitting weight decay to isolate the effect of the learning rate. We further formalize this insight in Theorem 3.10 where $y(t)y(t+1) \geq 0$ by characterizing a *stagnation region* around the origin where standard gradient updates become counterproductive.

**Theorem 3.10.** *Consider a continuously differentiable objective function $f : \mathbb{R}^d \to \mathbb{R}$. Assume that the coordinate of the function is bounded, namely, $\nabla[f(x)]_i \leq B$ for any coordinate $i$. Then $y(t)y(t+1) \geq 0$ if the following conditions hold:*

- *For adaptive learning rate with $M \neq 0, \epsilon = 0$,*

$$|y(t)| \leq U_1 \triangleq \left[\frac{1 - \lambda\eta_t}{\eta_t B}\right]^{1/(M-1)}.$$

- *For adaptive learning rate with $M = 0, \epsilon \neq 0$, if $\epsilon \leq \left[\frac{1-\eta_t\lambda}{\eta_t B}\frac{K-1}{K-2}\right]^{K-1}/(K-2)$,*

$$|y(t)| \leq U_2 \triangleq F^{-1}\left(\frac{\eta_t B}{1 - \lambda\eta_t}\right),$$

*where $F(U) = U + \epsilon U^{-(K-2)}$.*

- *Without adaptive learning rate,*

$$|y(t)| \leq U_3 \triangleq \left[\frac{1 - \lambda\eta_t}{\eta_t B}\right]^{1/(K-2)}.$$

Theorem 3.10 confirms that the reparameterization creates a region around the origin where the standard gradient update is ineffective, pulling the parameter towards the saddle point rather than pushing it across. The following Proposition 3.11 further compares the size of these regions.

**Proposition 3.11.** *Under the results in Theorem 3.10, assume that , it holds that*

1. *if $\frac{1-\lambda\eta_t}{\eta_t B} \leq 1$, it holds that $U_1 \leq U_3$;*
2. *if $\frac{1-\lambda\eta_t}{\eta_t B} \leq (1-\epsilon)^{(K-2)/(K-1)}$, it holds that $U_2 \leq U_3$;*
3. *if $\frac{1-\lambda\eta_t}{\eta_t B} \geq 1$, it holds that $U_1, U_2, U_3 > 1$.*

Proposition 3.11 demonstrates that an adaptive learning rate effectively rescales the optimization landscape. By narrowing the stagnation region, it ensures that the gradient signal points towards the true minimizer even when crossing the origin is required.

### 3.5. The advantages of $\ell_p$ regularizer

This section analyzes the advantages of $\ell_p$ regularizer over $\ell_1$ regularizer. We first show a concrete example in Example 3.12, verifying that under general cases, $\ell_1$ might not promote sparsity while $\ell_p$ still works.

**Example 3.12.** There exists a constraints $f$, constructing like $\|\theta - \theta_u\|_2^2 \leq (d-1)u^2$, where $\theta_u = (u, \cdots, u)$ and $u > 0$, such that

1. Its $\ell_0$ optimization leads to solution with $\|\theta_0^*\|_0 = 1$.
2. Its $\ell_1$ optimization leads to solution with $\|\theta_1^*\|_0 = d$, where $d$ denotes the dimension of $\theta$.
3. Its $\ell_p$ optimization leads to solution with $\|\theta_p^*\|_0 = 1$, for $p \leq 0.5$.

One could extend the insights in Example 3.12 into a general theorem, showing that [Cp] approximately returns sparse solutions. Before that, we first introduce the *Expansion Property* in Assumption 3.13.

**Assumption 3.13** (Expansion). Given any fixed $p \in (0, 1)$, assume that the objective function $f(x)$ is $(\mu, \beta)$-expansion. That is to say, for any $x$, there exists an expansion point $\hat{x}$ and two constants $\mu, \beta$, such that $\beta \geq p$, $\hat{x} = \arg\min_u f(u)$, and $\mu\|\hat{x} - x\|_p^\beta \leq f(x) - f(\hat{x})$.

The expansion assumption guarantees a favorable landscape around the minimizer manifold. Generally, it helps establish a connection between the solution of [Bp] and the minimizer of $f(x)$ with minimal $\ell_p$ norm ([Ap] in Equation 5). This assumption closely relates to the Quadratic Growth

Condition (Karimi et al., 2016), albeit being specifically defined within the realm of local minimizers. Subsequently, we present our theorem in Theorem 3.14.

**Theorem 3.14.** *Let $x_0$ denote the solution of [A0] in Equation 5 with $\|x_0\|_\infty \leq B$. Let $x^*$ denote the global minimizer of [Cp] with a bounded p-norm, given a fixed odd number $K$ and penalty $\lambda$. For a given constant $\gamma > 0$, denote $\bar{x}^* \triangleq \max\{x^*, \exp(-\gamma\sqrt{K/2})\}$ which eliminates the small values of $x^*$. If Assumption 3.13 holds with an expansion point having bounded p-norm, then if $K \geq \max\{2\gamma^2, B', 8(\frac{2B'+\gamma^2+2B'\gamma}{4B'+\gamma})^2\}$ where $B' = \max\{0, \log B\}$, as $\lambda \to 0$, it holds that*

$$\|\bar{x}^*\|_0 \leq (1 + (4B' + \gamma)\sqrt{\frac{2}{K}})\|x_0\|_0.$$

We next show in Corollary 3.15 how Theorem 3.14 leads to an exact recovery.

**Corollary 3.15** (Exact recovery). *Under the assumptions in Theorem 3.14, and assume that (a) $\gamma \geq \sqrt{2/K}\log M_K$, where $M_K$ denotes the minimal absolute value of $x^*$, and (b) $K > 2[(4B' + \gamma)\|x_0\|]^2$. Then it holds that $\bar{x}_0^* = x^*$.*

In Corollary 3.15, condition (a) guarantees that $\bar{x}^* = x^*$, and condition (b) guarantees that $\|\bar{x}^*\|_0 < \|x_0\|_0 + 1$. Notably, $M_p$ depends on both the parameter $K$ and the loss landscape, and therefore the above condition is problem-dependent, just as the R.I.P condition in compress sensing.

Besides, even if the exact recovery does not happen, one could derive that the elimination does not change the loss much. We next analyze the non-exact recovery case in Corollary 3.16 where $\bar{x}^* \neq x^*$.

**Corollary 3.16** (Non-exact recovery). *By eliminating elements smaller than $\exp(-\gamma\sqrt{K/2})$, the loss function $f(x)$ may increase while the sparsity improves. The Lipschitz condition helps control the loss. If $f$ is $L$-Lipschitz with respect to its 2-norm, namely, $|f(x_1) - f(x_2)| \leq L\|x_1 - x_2\|_2$, then the loss would increase at most $L\sqrt{d}\exp(-\gamma\sqrt{K/2})$ after elimination.*

*Remark* 3.17 (Why elimination). Elimination is necessary because small coordinates may make $p$-norm and 0-norm perform differently when the element depends on $p$. For example, consider a vector $u = (u, \cdots, u) \in \mathbb{R}^d$ for a pretty small constant $u$, *e.g.*, $u = \exp(-1/p^2)$. Then as $p \to 0$, it holds that $\|u\|_0 = d$ and $\|u\|_p^p = d\exp(-1/p) \to 0$. This phenomenon complicates the discussion, and therefore we apply elimination in Theorem 3.14.

*Remark* 3.18 (Comparison with compressed sensing). Although both compressed sensing and Theorem 3.14 focus on sparsity recovery tasks, they are fundamentally different. The compressed sensing approach encourages sparsity by leveraging the linear property and $\ell_1$ regularization. Dif-

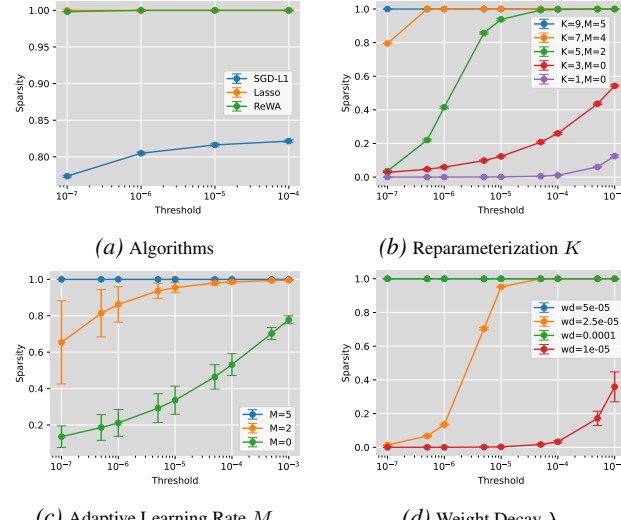

*(a)* Algorithms      *(b)* Reparameterization $K$

*(c)* Adaptive Learning Rate $M$      *(d)* Weight Decay $\lambda$

*Figure 2.* Performance of ReWA in linear regression regimes for different hyperparameters.

ferently, Theorem 3.14 does not require much problem information, and only relies on the property that $p$-norm is naturally close to 0-norm when $p$ is small. Unfortunately, Theorem 3.14 cannot directly recover the compressed sensing results by setting $p = 1$.

## 4. Experiments

In this section, we conduct experiments on both synthetic and real-world datasets to evaluate the effectiveness of ReWA. We first show the empirical results on linear synthetic datasets in Section 4.1 and CIFAR-10 / ImageNet in Section 4.2, showing the effectiveness of ReWA.

### 4.1. Experiments on Synthetic Dataset

**Datasets and Models.** The synthetic dataset exhibits a linear regime with added noise, and is fitted with linear models. This dataset is used to simulate the signal recovery. Precisely, the data are generated based on a linear ground truth $y = x^\top \beta^* + \epsilon$, where $x, \beta^* \in \mathbb{R}^{10000}, y, \epsilon \in \mathbb{R}$. The ground truth $\beta^*$ is sparse with a single nonzero entry, directly matching the exact recovery setting of Corollary 3.15. The noise $\epsilon$ follows a normal distribution $\mathcal{N}(0, 1)$, resulting in a Bayesian optimal of 1. A linear model is applied to fit a training dataset that contains $n = 2000$ samples over 800 epochs. The base algorithm utilized is SGD with a batchsize $b = 25$ and a learning rate $\lambda = 2 \times 10^{-4}$.

**Evaluation Metrics.** We consider both the sparsity metric, where the sparsity is evaluated as the number of weights below several given thresholds. Across all experiments, the test loss consistently approaches the Bayesian optimal.

**Baseline Algorithms and Ablation Study.** We compare our methods with the following baselines: *SGD-L1* (SGD with $\ell_1$ regularization which is a commonly-used technique), *LASSO* (LASSO with coordinate-wise gradient descent). We also consider the following ablation studies: ReWA with different Reparameterization $K$, ReWA with different Adaptive Learning Rate $M$, and ReWA with different weight decay $\lambda$. Notably, the comparison to PowerPropogation (Schwarz et al., 2021) falls in the third branch.

**Results and Discussions.** The results on the sparsity with different methods are illustrated in Figure 2. A well-performed sparse training method is expected to return solutions with sparsity given different thresholds. Figure 2 demonstrates that ReWA works comparably with LASSO under linear regimes, both outperforming SGD with $\ell_1$ penalty. This demonstrates the effectiveness of ReWA even under simple linear regimes, and we remark that ReWA performs better in non-linear regimes (see Section 4.2). For the ablation studies, we find that (a) larger $K$ with corresponding larger $M$ may lead to sparser solutions; (b) larger $M$ may lead to sparser solutions given a hyperparameter $K$; and (c) larger weight decay $\lambda$ may lead to sparser solutions. These ablation studies demonstrate that reparameterization, weight decay, and adaptive learning rate are indeed crucial factors in ReWA.

### 4.2. Experiments on Real-world Dataset

**Datasets and Models.** The datasets includes (a) CIFAR-10 with ResNet-18; and (b) ImageNet with ResNet-50.

**Evaluation Metrics.** We consider how the model maintains accuracy given the compression ratio (Tanaka et al., 2020), following Liu & Wang (2023). Here the compression ratio is defined as the number of parameters divided by the number of non-zero parameters after the pruning.

**Baseline Algorithms and Ablation Study.** For CIFAR-10, the red line represents the optimal performance of ReWA (parameterized by $M$). The baselines include: (a) $\ell_1$, which directly minimizes the loss with $\ell_1$ penalty using SGD; (2) Spred Liu & Wang (2023), which is related to ReWA with $K = 2$. For ImageNet, the brown line represents the optimal performance returned by ReWA (parameterized by $\epsilon$). The baselines include: (a) $\ell_1$; (b) directly deploying SGD on the raw loss. Besides, we conduct ablation studies on (a) ReWA without weight decay and adaptive learning rate; (b) ReWA with only adaptive learning rate $\epsilon$; (c) ReWA with only weight decay. We remark that $\ell_{1/2}$ (Xu et al., 2012) penalties are special cases of the $\ell_p$ family covered by our framework, so we do not treat them as separate baselines.

**Results and Discussions.** The results are illustrated in Figure 3. For CIFAR-10, ReWA maintains competitive accuracy until the compression ratio exceeds $10^2$, whereas prior

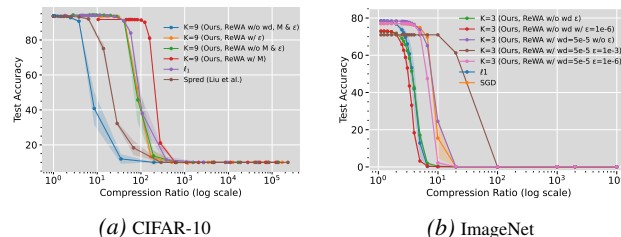

*(a)* CIFAR-10      *(b)* ImageNet

*Figure 3.* Performance of ReWA in real-world datasets.

methods on the same ResNet-18 architecture typically exhibit significant accuracy degradation once the compression ratio reaches $10^2$ (see Figure 6 of Tanaka et al. (2020)). For ImageNet, ReWA similarly maintains accuracy at relatively high compression ratios.

We provide ablation studies, computational overhead, and compatibility with AdamW in Appendix D.

## 5. Conclusion

The paper presents ReWA, a novel approach to sparse optimization that addresses the limitations of traditional $\ell_1$ and $\ell_p$ regularization methods. By integrating reparameterization, weight decay, and an adaptive learning rate, ReWA achieves improved sparsity without significantly sacrificing accuracy, as demonstrated through experiments on CIFAR-10 and ImageNet. Theoretical analysis supports the practicality and stability of ReWA, showing its close relation to $\ell_p$ regularization with $0 < p < 1$.

**Limitations and future work.** The theoretical analysis relies on standard assumptions (smoothness, subspace, and expansion conditions) that are difficult to verify directly for deep networks in practice. The method introduces hyper-parameters $K$, $M$, and $\varepsilon$; while we provide practical guidelines, automated selection remains future work. The current experiments focus on image classification with ResNet architectures, which is the standard benchmark in the sparse optimization literature. Extending ReWA to text and language modeling tasks with Transformer-based architectures is a natural next step; Algorithm 2 provides a ReWA+AdamW variant designed for this setting. We leave a thorough empirical evaluation on NLP benchmarks to future work.

## Acknowledgements

This work is supported by Shanghai Science and Technology Development Funds 24YF2711700 and Fundamental Research Funds for the Central Universities 2024110586.

## Impact Statement

This paper presents a theoretical analysis of sparse optimization. It involves no human subjects, data, or model release, and we are not aware of societal consequences that require specific highlighting here.

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

# Appendix

We include the omitted algorithms and illustrations here. Section A provides additional related works. Section B provides a detailed comparison with SCAD, MCP, and adaptive Lasso. Section C presents the omitted proofs. Section D provides additional experimental details, including ablation studies (Section D.5) and computational overhead (Section D.6).

## A. Related Work (Extended)

**Reparameterization with weight decay leads to sparsity.** The evidence that reparameterization with weight decay may lead to sparsity can be dated back to Grandvalet (1998), where they prove that such technique induces an $\ell_1$ regularization in linear regimes (Li et al., 2021a; 2023). A line of work extends from linear regimes to general function classes, *e.g.*, Poon & Peyré (2021; 2023) analyzes realistic algorithms under convex, proper, semicontinuous functions, and Liu & Wang (2023) further deploy such techniques into deep learning regimes. However, when extending the aforementioned $\ell_1$ related reparameterization into $\ell_p$ related reparameterization, the computational instability emerges. This branch of extension is pioneered by Hoff (2017), where they derive the reparameterization for $\ell_p$ regularization under general function classes, but do not propose how it works in iterative gradient methods. Later, Kolb et al. (2023) generalizes the method to SGD regimes, and conducts experiments on synthetic datasets. However, *deploying the aforementioned techniques in complex real-world datasets remains challenging, as high reparameterization levels cause computational instability.*

Another line of work tries to understand specific model structures by regarding them as a type of reparameterization. For example, matrix factorization tasks naturally have a reparameterization-like term (Srebro et al., 2004; Rennie & Srebro, 2005; Hastie et al., 2015) and multi-layer neural networks can be regarded as a type of reparameterization (Neyshabur et al., 2015; Savarese et al., 2019; Ongie et al., 2020; Pilanci & Ergen, 2020; Parhi & Nowak, 2021; Dai et al., 2021; Tibshirani, 2021; Yang et al., 2022; Jagadeesan et al., 2022; Jacot et al., 2022; Parhi & Nowak, 2023).

**Reparameterization without weight decay also benefits sparsity.** Reparameterization closely relates to the sparsity even without weight decay (Amid & Warmuth, 2020; Li et al., 2022; Liu et al., 2022). From the theoretical perspective, existing works have shown the implicit bias of reparameterization under specific loss forms, *e.g.*, linear regimes (Gunasekar et al., 2018; Vaskevicius et al., 2019; Woodworth et al., 2020; Gissin et al., 2020; Chou et al., 2021; Wu & Rebeschini, 2021; Li et al., 2021b; Zhao et al., 2022; Pillaud-Vivien et al., 2022; Ma & Fattahi, 2022), matrix factorization (Arora et al., 2019; You et al., 2020), and single-index models (Fan et al., 2023).

*Discussion: relation to the mirror-flow line of work.* A complementary line of work analyzes reparameterization-based sparsification via the mirror-flow framework of Li et al. (2022), under which gradient flow on $\boldsymbol{x} = g(\boldsymbol{w})$ is equivalent to a mirror flow with a Bregman potential determined by $g$. Building on this, Jacobs & Burkholz (2025) study the $\boldsymbol{m} \odot \boldsymbol{w}$ parameterization ($K = 2$) with time-varying weight decay and obtain a time-dependent Bregman potential interpolating between implicit $\ell_2$ and $\ell_1$; Gadhikar et al. (2026) use the same parameterization to enable sign flips; and Jacobs et al. (2025) characterize how explicit regularization reshapes the implicit bias.

This framework is tied to $K = 2$: its mirror-flow equivalence requires a commuting condition that, as shown in Theorem D.2 of Jacobs et al. (2025), fails for the $K$-fold product with weight decay when $K > 2$. In the $\boldsymbol{x}$-domain ($\epsilon = 0$), our ReWA iteration reads

$$\boldsymbol{x}(t+1) = \boldsymbol{x}(t) \odot \left[ (1 - \lambda \eta_t)\mathbf{1} - \eta_t \, \boldsymbol{x}^{(M-1)/K}(t) \odot \nabla f(\boldsymbol{x}(t)) \right]^K .$$

The coefficient $\boldsymbol{x}^{(M-1)/K}$ is nonzero at the origin only at $M = 1$, giving $p = 1 + (1 - M)/K = 1$ ($\ell_1$), which in the non-adaptive setting is the $K = M + 1 = 2$ case covered by the mirror-flow framework and offers little benefit. In the regime of interest, $p < 1$ ($M > 1$), the coefficient vanishes at the origin (a high-order saddle), where a continuous flow would stall. The discrete dynamics can nonetheless cross zero: near the origin, a large learning rate drives the per-coordinate factor below $-1$, and raising it to the $K$-th power ($K$ large and odd) yields a large sign-flipping step across zero, accelerated by larger $K$. Hence a large early learning rate enables exploration through zero crossings (and sign correction), while a small late learning rate keeps the factor within $(-1, 1)$ and lets the iterate settle. This is consistent with the observation in Gadhikar et al. (2026) that early sign flips are crucial.

In contrast to the $\boldsymbol{m} \odot \boldsymbol{w}$ line, we keep the factors tied ($\boldsymbol{x} = \boldsymbol{y}^K$), operate in discrete time, and target $\ell_p$ with $p < 1$ rather than only $\ell_1$. Our analysis thus offers a complementary perspective that reaches the $K > 2$ regime lying outside the mirror-flow framework.

$\ell_p$ **regularization.** The reparameterized form is derived from $\ell_p$ regularization tasks (Frank & Friedman, 1993; Fu, 1998). Compared to $\ell_1$ regularization, $\ell_p$ regularization is more robust and stable (Chartrand, 2007; Chartrand & Staneva, 2008; Saab et al., 2008; Chartrand, 2009; Zheng et al., 2017). As $p$ goes to zero, $\ell_p$-minimization naturally converge to $\ell_0$ optimization. Unfortunately, although $\ell_p$-minimization is solvable in linear regimes (Marjanovic & Solo, 2012; 2013; Wen et al., 2018), optimizing it in general regimes is hard due to its unbounded gradient and non-smooth nature. Existing approaches usually require approximation on loss (Fan & Li, 2001; Hunter & Li, 2005), or lack theoretical backbone (Chartrand, 2007).

**Other sparse optimization techniques.** The most popular technique is $\ell_1$ regularization (Zhao, 2018), *e.g.*, compressed sensing (Candes & Tao, 2006; Candes, 2008; Mousavi et al., 2019) which derives that $\ell_1$-approximation exactly recovers the signal under noiseless linear regimes. However, $\ell_1$-approximation may not be the best choice in general regimes due to its bias (Zhang & Huang, 2008; Huang et al., 2022). Besides $\ell_1$ and $\ell_p$ regularization, other techniques include (a) directly solving $\ell_0$ regularization using combinatorial tricks (Portilla & Mancera, 2007; Feng et al., 2013), and (b) smoothing the $\ell_0$ regularization problems (Louizos et al., 2018).

**Applications of sparsity.** Sparsity is among the most fundamental factors in applications, which can be dated back to variable selection (Bradley & Mangasarian, 1998; George, 2000) and signal processing (Lustig et al., 2007; 2008; Duarte-Carvajalino & Sapiro, 2009; Marvasti et al., 2012; Stanković et al., 2019). In modern machine learning regimes, the most direct application is network pruning, which is an approach to reducing the computational and memory cost for neural networks (Han et al., 2016; 2015; Li et al., 2017; Louizos et al., 2018; Savarese et al., 2020; Xia et al., 2023). Researchers also find the utility of sparsity in continual learning (Schwarz et al., 2021; Kang et al., 2022), fine-tuning (Panigrahi et al., 2023), and out-of-distribution (Zhang et al., 2021; Sun & Li, 2022).

**Adaptive learning rate methods.** Adaptive optimizers such as Adam (Kingma, 2014) and AdamW (Loshchilov & Hutter, 2019) scale each parameter's update by the square root of its accumulated squared gradients, enabling robust optimization across diverse loss landscapes (Kingma, 2014). In the context of high-order reparameterization $\boldsymbol{x} = \boldsymbol{y}^K$, the chain rule introduces a multiplicative factor $K\boldsymbol{y}^{K-1}$, which shrinks the effective gradient dramatically when $\boldsymbol{y}$ is near zero. Adam's per-coordinate normalization implicitly cancels this factor, providing a principled explanation for why adaptive methods are preferable over plain SGD in reparameterized sparse training. AdamW (Loshchilov & Hutter, 2019) further decouples weight decay from the gradient update, aligning naturally with the structure of ReWA where the regularization term and the objective gradient are handled separately. However, standard Adam corresponds only to the special case $M = 0, \epsilon \neq 0$ of ReWA; our explicit $M$ parameter provides finer control over the sparsity-inducing regularization strength, which is not available in off-the-shelf adaptive optimizers.

## B. Comparison with SCAD, MCP, and Adaptive Lasso

This section elaborates on the relationship between $\ell_p$ regularization and other nonconvex methods such as SCAD, MCP, and adaptive Lasso. Our claim is not that $\ell_p$ dominates these methods, but that they offer **complementary strengths** and that $\ell_p$ is independently important as a research direction. As illustrated in Figure 4, there exist regimes where $\ell_p$ correctly identifies sparse structure where $\ell_1$ (and methods with $\ell_1$-like local behavior) may not. We support this via three arguments below.

**Long-standing independent interest.** $\ell_p$ regularization ($p < 1$) has a rich and independent theoretical history: Chartrand (2007) proved exact sparse recovery via $\ell_p$; Chartrand & Staneva (2008) derived RIP conditions for $\ell_p$; Xu et al. (2012) developed thresholding theory for $\ell_{1/2}$ showing its superiority over $\ell_1$; Zheng et al. (2017) established conditions where $\ell_p$ strictly outperforms $\ell_1$; Kolb et al. (2023) extended reparameterization-based $\ell_p$ to SGD. This body of work shows $\ell_p$ is a distinct and well-studied research direction, not merely an alternative to SCAD/MCP.

**Direct connection to $\ell_0$.** As $p \to 0$, $\ell_p$ approaches $\ell_0$. This connection provides geometric insights fundamentally different from those offered by SCAD/MCP, whose design targets unbiasedness for large coefficients rather than approximating $\ell_0$.

**Complementary empirical and geometric value.** **Empirical evidence.** Bui et al. (2021) treats $\ell_p$, $T\ell_1$, MCP, and SCAD as parallel important sparse penalties, finding complementary behavior rather than single-method dominance. Concretely:

- $\ell_p$ *yields significantly higher compression*: it generally prunes more parameters and FLOPs than $\ell_1$, MCP, and SCAD. In high-sparsity regimes, $\ell_p$ models remain functional where SCAD/MCP become over-pruned.
- *SCAD/MCP yield better post-retraining accuracy* at similar compression levels to $\ell_1$. This reflects their oracle property

---

**Algorithm 2** ReWA with AdamW

---

**Require:** object function $f(\boldsymbol{x})$ with gradient $\nabla f(\boldsymbol{x})$, initialization $\boldsymbol{x}(0)$, weight decay parameter $\lambda$, learning rate scheduler $\eta_t \in \mathbb{R}$, element-wise power function $(\cdot)^K$.

1: Set $\boldsymbol{y}(0) = \boldsymbol{x}(0)^{1/K}$
2: Initialize $Adam$ optimizer using $y(0)$ as parameters
3: **for** $t \in [T]$ **do**
4:     Calculate $\nabla f(\boldsymbol{y}^K(t))$
5:     Replace gradient of $\boldsymbol{y}(t)$ in $Adam$ optimizer with $\eta_t \frac{\boldsymbol{y}^M(t)}{\boldsymbol{y}^{K-1}(t)+\epsilon\boldsymbol{1}} \odot \boldsymbol{y}^{K-1}(t) \odot \nabla f(\boldsymbol{y}^K(t))$
6:     Optimize $\boldsymbol{y}(t)$ using $Adam$ optimizer with replaced gradient
7:     (Optional) $\boldsymbol{x}(t+1) = \boldsymbol{y}^K(t+1)$
8: **end for**
**Ensure:** $\boldsymbol{x}(T) = \boldsymbol{y}^K(T)$ .

---

(Fan & Li, 2001): eliminating shrinkage bias for large coefficients ensures unbiased estimation for surviving weights, effectively preserving model capacity.

Thus, $\ell_p$ excels at achieving high sparsity, while SCAD/MCP excel at accuracy preservation under moderate pruning.

**Connection to ReWA.** Bui et al. (2021) optimized $\ell_p$ via subgradient descent, which suffers from unbounded gradients near zero, causing pruning instability. ReWA addresses this: reparameterization yields bounded gradients around zero (Section 3.1.1), and the adaptive learning rate stabilizes training (Example 3.2). This retains $\ell_p$'s compression advantage while mitigating the instability of direct subgradient optimization. ReWA matches SGD in runtime and memory (Table 2). Since prox-SCAD/MCP likewise require an extra element-wise step, they offer no complexity advantage over ReWA. In this sense, ReWA is to $\ell_p$ what proximal operators are to SCAD/MCP, which transforming a challenging penalty into a reliable optimization method.

**Geometric analysis.** We next provide a geometric explanation for the complementarity. By definition, $\mathrm{SCAD}(\theta_i) \equiv \lambda|\theta_i|$ for all $|\theta_i| \leq \lambda$. Thus, for any fixed $\lambda$, SCAD inherits the exact local shrinkage behavior of $\ell_1$ within this region. Its oracle property is an asymptotic guarantee that activates only for large coefficients reaching the unbiased tail ($|\theta_i| > a\lambda$). Consequently, under finite-sample, weak-signal, or heterogeneous-scale settings, SCAD behaves locally like $\ell_1$ for a substantial subset of weights. In contrast, $\ell_p$ ($p < 1$) maintains a sharp singularity at zero at all scales, inducing stronger penalization on small coefficients. This fundamental geometric difference explains why $\ell_p$ inherently favors higher sparsity even when SCAD's asymptotic conditions are unmet.

To illustrate, consider a two-dimensional constrained analysis with $(\theta_0 - r)^2 + (\theta_1 - r)^2 = r^2$ (a circle tangent to both axes at the origin, parametrized by signal strength $r$; see Figure 4 for an illustration of the sparse vs. dense solution geometry). This is analogous to Example 3.12, which already establishes that $\ell_p$ ($p \leq 0.5$) favors sparse solutions in settings where $\ell_1$ does not. For SCAD, the analysis extends as follows:

- $\ell_p$ consistently favors sparse solutions (for $p < 0.564$), regardless of the signal strength $r$.
- SCAD yields dense solutions when the signal $r$ falls below a critical threshold $r_c$. Crucially, $r_c \geq 2.56\lambda$ for all $a > 2$ (e.g., $r_c \approx 4.03\lambda$ at $a = 3.7$); no choice of $a$ eliminates this dense-preference regime, as coefficients remain in the $\ell_1$-like region.

**Summary.** $\ell_p$'s scale-invariant singularity inherently enforces sparsity even for moderate or weak signals (enabling high compression). While SCAD preserves large weights via its oracle property but relies on $\ell_1$-like shrinkage for smaller ones, $\ell_p$ ($p < 1$) induces scale-adaptive shrinkage (via $|\theta|^{p-1}$), driving stronger sparsification on small coefficients, while its bias on larger weights vanishes as $p \to 0$. ReWA stabilizes this effect during optimization. Both approaches offer distinct, non-redundant value, motivating the critical need for reliable optimizers such as ReWA.

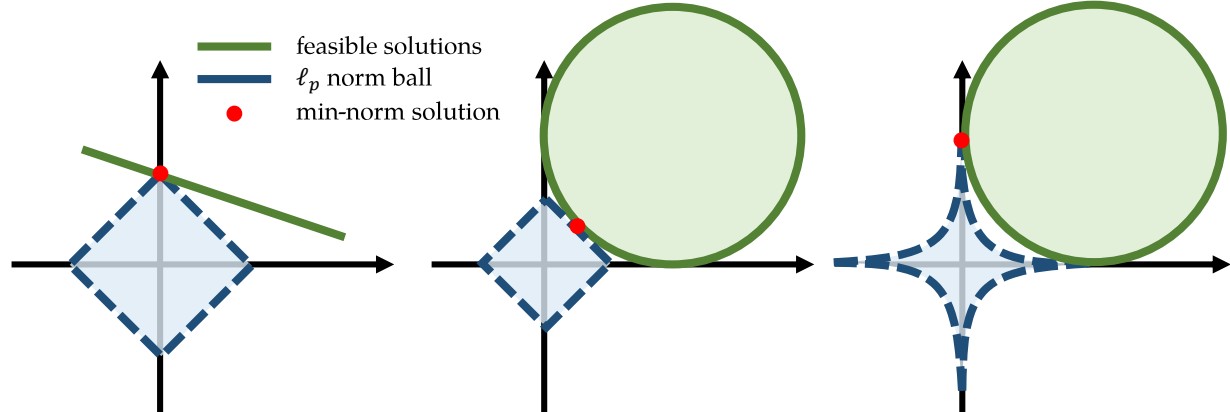

*Figure 4.* $\ell_1$ regularization fails in general cases. The solution is sparse if it falls in the coordinate.
Left: $\ell_1$ regularization returns sparse solutions in linear cases (compressed sensing).
Middle: $\ell_1$ regularization returns non-sparse solutions in general non-linear cases.
Right: $\ell_p$ regularization ($p \in (0, 1)$) may perform better and return sparse solutions in general non-linear cases.

# C. Omitted Proofs

This section provides omitted proofs. Section C.1 provides necessary preliminaries and additional notations. Section C.2 provides proofs of Lemma 3.1. Section C.3 provides proofs of Example 3.2. Section C.4 provides proofs of Theorem 3.3. Section C.5 provides proofs of Theorem 3.7. Section C.6 provides proofs of Example 3.8. Section C.7 provides proofs of Theorem 3.9. Section C.8 provides proofs of Theorem 3.10. Section C.9 provides proofs of Proposition 3.11. Section C.10 provides proofs of Example 3.12. Section C.11 provides proofs of Theorem 3.14.

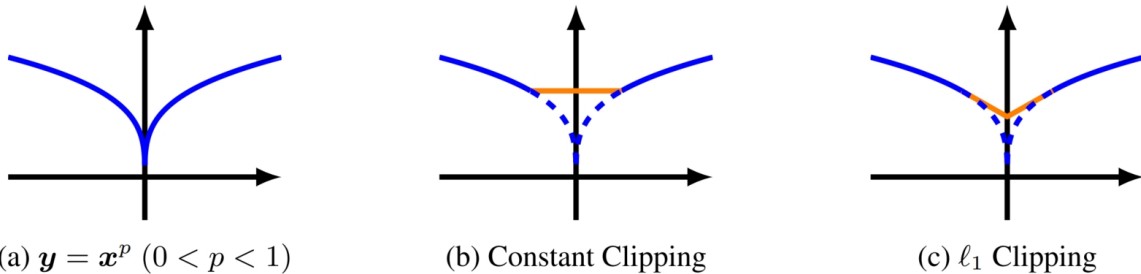

(a) $y = x^p$ $(0 < p < 1)$      (b) Constant Clipping      (c) $\ell_1$ Clipping

*Figure 5.* Clipping for $\ell_p$ regularization. The blue lines (solid and dotted) represent the original $\ell_p$ regularization, and the orange lines represent the clipped version at some clipping point.

## C.1. Preliminaries

We first present preliminaries before the proof, including additional notations and basic knowledge about compressed sensing. We would start from the notations on $\ell_0$ minimization.

Sparse optimization problems can be formally represented through $\ell_0$ minimization ([A0] in Equation 5). However, directly solving $\ell_0$ minimization is known to be NP-hard (Natarajan, 1995a; Chen et al., 2017). To address this challenge, prior works propose two relaxations: $\ell_1$ minimization ([A1] in Equation (5)) and $\ell_p$ minimization ($p \in (0,1)$, [Ap] in Equation (5)).

$$
\begin{aligned}
[A0] &: \min_{\boldsymbol{x}} \|\boldsymbol{x}\|_0 \quad \text{subject to} \quad \boldsymbol{x} \in \arg\min_{\boldsymbol{u}} f(\boldsymbol{u}), \\
[A1] &: \min_{\boldsymbol{x}} \|\boldsymbol{x}\|_1 \quad \text{subject to} \quad \boldsymbol{x} \in \arg\min_{\boldsymbol{u}} f(\boldsymbol{u}), \\
[Ap] &: \min_{\boldsymbol{x}} \|\boldsymbol{x}\|_p \quad \text{subject to} \quad \boldsymbol{x} \in \arg\min_{\boldsymbol{u}} f(\boldsymbol{u}),
\end{aligned}
\tag{5}
$$

where $\boldsymbol{x} \in \mathbb{R}^d$ denotes the signal, and $f(\boldsymbol{x}) \in \mathbb{R}^m$ denotes the constraints. One of the most famous $\ell_1$ relaxations is *compressed sensing* (Candes & Tao, 2006) illustrated in Example C.1.

**Example C.1** (Compressed Sensing). Compressed sensing tasks consider the constraints $f(\boldsymbol{x}) = \|A\boldsymbol{x} - \boldsymbol{y}\|^2$, where $A \in \mathbb{R}^{m \times d}$ denotes the measurement matrix and $\boldsymbol{y}$ denotes the corresponding observation. In compressed sensing, the [A0] solution can be exactly recovered by solving [A1] under noiseless linear regimes. However, [A1] might not always be the most suitable choice, particularly for non-linear problems. We refer to Figure 4 for further illustration.

For the three problems in Equation 5, we also focus on their corresponding optimization formulations in Equation 6. In general, [B1] can be efficiently optimized while [B0] and [Bp] cannot, since [B0] is non-continuous and [Bp] is non-smooth with unbounded gradient around zero.

$$
\begin{aligned}
[B0] &: \min_{\boldsymbol{x}} L_0(\boldsymbol{x}) = f(\boldsymbol{x}) + \lambda \|\boldsymbol{x}\|_0, \\
[B1] &: \min_{\boldsymbol{x}} L_1(\boldsymbol{x}) = f(\boldsymbol{x}) + \lambda \|\boldsymbol{x}\|_1, \\
[Bp] &: \min_{\boldsymbol{x}} L_p(\boldsymbol{x}) = f(\boldsymbol{x}) + \lambda \|\boldsymbol{x}\|_p^p.
\end{aligned}
\tag{6}
$$

In this paper, we focus on a different optimization form presented in Equation (7), which reparameterizes the vector $\boldsymbol{x}$ with

$\boldsymbol{y}_1 \odot \boldsymbol{y}_2 \odot \cdots \odot \boldsymbol{y}_K$ where $\odot$ denotes the element-wise product:

$$[\text{Cp}] : \min_{\boldsymbol{y}} L'_p(\boldsymbol{y}) = f(\boldsymbol{y}_1 \odot \boldsymbol{y}_2 \odot \cdots \odot \boldsymbol{y}_K) + \lambda \sum_{i \in [K]} \|\boldsymbol{y}_i\|_2^2. \tag{7}$$

We set $K$ as an odd number for simplicity.

## C.2. Proof of Lemma 3.1

We prove the three arguments separately. The last two arguments are inspired by Kolb et al. (2023).

**Stationary Points.** Let $\tilde{\boldsymbol{y}}_i, i \in [K]$ denote the stationary points of [Cp]. Firstly, we would like to show that $\tilde{\boldsymbol{y}}_1 \odot \tilde{\boldsymbol{y}}_1 = \tilde{\boldsymbol{y}}_2 \odot \tilde{\boldsymbol{y}}_2 = \cdots = \tilde{\boldsymbol{y}}_K \odot \tilde{\boldsymbol{y}}_K$. Notice that for each $\tilde{\boldsymbol{y}}_i$, the gradient of [Cp] satisfies that

$$\odot_{j \neq i} \tilde{\boldsymbol{y}}_j \nabla f(\tilde{\boldsymbol{y}}_1 \odot \cdots \odot \tilde{\boldsymbol{y}}_K) + 2\lambda \tilde{\boldsymbol{y}}_i = \boldsymbol{0}. \tag{8}$$

Therefore, for each $i$, it holds that

$$\tilde{\boldsymbol{y}}_1 \odot \cdots \odot \tilde{\boldsymbol{y}}_K \nabla f(\tilde{\boldsymbol{y}}_1 \odot \cdots \odot \tilde{\boldsymbol{y}}_K) = -2\lambda \tilde{\boldsymbol{y}}_i \odot \tilde{\boldsymbol{y}}_i,$$

and therefore $\tilde{\boldsymbol{y}}_i \odot \tilde{\boldsymbol{y}}_i = \tilde{\boldsymbol{y}}_j \odot \tilde{\boldsymbol{y}}_j$ for any $i, j \in [K]$.

Let $\tilde{\boldsymbol{y}} \in \mathbb{R}^d$ denote the vector whose absolute value is equal to that of $\tilde{\boldsymbol{y}}_i$, and sign the same as $\odot \tilde{\boldsymbol{y}}_i = \tilde{\boldsymbol{y}}_1 \odot \cdots \odot \tilde{\boldsymbol{y}}_K$. Note that by denoting $\tilde{\boldsymbol{x}} = \odot \tilde{\boldsymbol{y}}_i$, it holds that $\tilde{\boldsymbol{x}} = \tilde{\boldsymbol{y}}^K$, where the power denotes the element-wise operation. We can then rewrite the gradient of [Cp] as

$$\begin{aligned}
&\odot_{j \neq i} \tilde{\boldsymbol{y}}_j \nabla f(\tilde{\boldsymbol{y}}_1 \odot \cdots \odot \tilde{\boldsymbol{y}}_K) + 2\lambda \tilde{\boldsymbol{y}}_i \\
&= \odot_{j \neq i} \tilde{\boldsymbol{y}}_j [\nabla f(\tilde{\boldsymbol{y}}_1 \odot \cdots \odot \tilde{\boldsymbol{y}}_K) + 2\lambda \tilde{\boldsymbol{y}}^{2-K}] \\
&= \odot_{j \neq i} \tilde{\boldsymbol{y}}_j [\nabla f(\tilde{\boldsymbol{x}}) + 2\lambda \tilde{\boldsymbol{x}}^{(2-K)/K}] \\
&= \boldsymbol{0}.
\end{aligned}$$

By denoting $p = 2/K$ and adjusting the sign, the above equation is equivalent to

$$\tilde{\boldsymbol{x}}[\nabla f(\tilde{\boldsymbol{x}}) + 2\lambda \tilde{\boldsymbol{x}}^{p-1}] = \boldsymbol{0}, \tag{9}$$

where we use the fact that $\odot_{j \neq i} \tilde{\boldsymbol{y}}_j = \boldsymbol{0}$ iff $\tilde{\boldsymbol{x}} = \boldsymbol{0}$ for each coordinate.

Note that the gradient of [Bp] is

$$\nabla f(\tilde{\boldsymbol{x}}) + pK\lambda \tilde{\boldsymbol{x}}^{p-1}. \tag{10}$$

Therefore, the stationary points of [Cp] is the $V$-substationary points of [Bp], with the parameter $K\lambda = \frac{2}{p}\lambda$.

**Local Minimizer.** Different from the analysis in stationary points, here we do not require that the function is differentiable. Still, we first prove by contradiction that the local minimizer satisfies that

$$|\tilde{\boldsymbol{y}}_1| = |\tilde{\boldsymbol{y}}_2| = \cdots = |\tilde{\boldsymbol{y}}_K|,$$

where the absolute operation is element-wise. We first assume without loss of generality that there exists a local minimizer $\bar{\boldsymbol{y}}_1, \bar{\boldsymbol{y}}_2, \cdots, \bar{\boldsymbol{y}}_K$, such that $\bar{\boldsymbol{y}}_1^{(1)} < \frac{1}{\sqrt{1+\delta}} \bar{\boldsymbol{y}}_2^{(1)}$, where $\cdot^{(i)}$ denotes the $i$-th coordinate and $\delta$ denotes a sufficiently small constant. Then we conduct a small perturbation $\bar{\boldsymbol{y}}'$ around $\bar{\boldsymbol{y}}$, where

$$\bar{\boldsymbol{y}}_1^{(1)\prime} = \sqrt{1+\delta'}\bar{\boldsymbol{y}}_1$$
$$\bar{\boldsymbol{y}}_2^{(1)\prime} = \frac{1}{\sqrt{1+\delta'}}\bar{\boldsymbol{y}}_2,$$

for some small enough $\delta' < \delta$ and the other elements keep remained. It is obviously that the distance of $\bar{\boldsymbol{y}}'$ and $\bar{\boldsymbol{y}}$ can always be bounded for a small $\delta'$. Then it holds that

$$f(\bar{\boldsymbol{y}}'_1 \odot \bar{\boldsymbol{y}}'_2 \odot \cdots \odot \bar{\boldsymbol{y}}_K) = f(\bar{\boldsymbol{y}}_1 \odot \bar{\boldsymbol{y}}_2 \odot \cdots \odot \bar{\boldsymbol{y}}_K),$$
$$\left\|\bar{\boldsymbol{y}}_1^{(1)\prime}\right\|^2 + \left\|\bar{\boldsymbol{y}}_2^{(1)\prime}\right\|^2 = (1+\delta')\|\bar{\boldsymbol{y}}_1^{(1)}\|^2 + \frac{1}{1+\delta'}\|\bar{\boldsymbol{y}}_2^{(1)}\|^2 < \|\bar{\boldsymbol{y}}_1^{(1)}\|^2 + \|\bar{\boldsymbol{y}}_2^{(1)}\|^2,$$

where the last equation can be verified under the condition that $\bar{\boldsymbol{y}}_1^{(1)} < \frac{1}{\sqrt{1+\delta}}\bar{\boldsymbol{y}}_2^{(1)}$ and $\delta' < \delta$. That is to say, perturbing $\bar{\boldsymbol{y}}$ to $\bar{\boldsymbol{y}}'$ would decrease the loss function, which contradicts the requirement that $\bar{\boldsymbol{y}}$ is a local minimizer. Therefore, the local minimizer must satisfy that $|\tilde{\boldsymbol{y}}_1| = |\tilde{\boldsymbol{y}}_2| = \cdots = |\tilde{\boldsymbol{y}}_K|$.

We next only consider the case with $|\tilde{\boldsymbol{y}}_1| = |\tilde{\boldsymbol{y}}_2| = \cdots = |\tilde{\boldsymbol{y}}_K|$, and rewrite $\tilde{\boldsymbol{x}} = \odot\tilde{\boldsymbol{y}}_i$. In such a case, $\|\tilde{\boldsymbol{y}}_i\|^2 = \|\tilde{\boldsymbol{x}}\|^{2/K}$. Notably, one can always rewrite [Bp] using the form of [Cp] and restricting it under the case $|\tilde{\boldsymbol{y}}_1| = |\tilde{\boldsymbol{y}}_2| = \cdots = |\tilde{\boldsymbol{y}}_K|$.

In the [Cp] problem, the point $\tilde{\boldsymbol{y}}$ is a local minimizer means that there exists a ball $\mathcal{B}_\delta(\tilde{\boldsymbol{y}})$ with radius $\delta$ such that

$$\mathcal{L}'_p(\tilde{\boldsymbol{y}}) \leq \min_{\boldsymbol{y} \in \mathcal{B}_\delta(\tilde{\boldsymbol{y}})} \mathcal{L}'_p(\boldsymbol{y}).$$

Note that the relationship between $\boldsymbol{x}$ and $\boldsymbol{y}$ is monotone, and therefore the small perturbation of $\boldsymbol{y}$ covers a perturbation of $\boldsymbol{x}$ (although with a different radius). Therefore, the corresponding $\tilde{\boldsymbol{x}} = \tilde{\boldsymbol{y}}^K$ is also a local minimizer.

**Global Minimizer.** From previous analysis on local minimizer, it holds that $|\tilde{\boldsymbol{y}}_1| = |\tilde{\boldsymbol{y}}_2| = \cdots = |\tilde{\boldsymbol{y}}_K|$. and we can rewrite [Cp] as the form in [Bp]. Notice that for every parameter $\boldsymbol{x}$, it can be represented as the form that satisfies $|\tilde{\boldsymbol{y}}_1| = |\tilde{\boldsymbol{y}}_2| = \cdots = |\tilde{\boldsymbol{y}}_K|$. Therefore, the global minimizer of [Cp] corresponds to the global minimizer of [Bp].

### C.3. Proof of Example 3.2

**Constant learning rate.** Recall the iteration

$$\boldsymbol{y}(t+1) = \boldsymbol{y}(t) - 2\eta'\boldsymbol{y}^{K-1}(t)(\boldsymbol{y}^K(t) - 1). \tag{11}$$

We next use deduction to prove that $\boldsymbol{y}(t) \in [-1, 0]$ for all $T$.

Firstly, when $T = 0$, $\boldsymbol{y}(t) \in [-1, 0]$ directly holds.

We next assume that $\boldsymbol{y}(t) \in [-1, 0]$ and prove that $\boldsymbol{y}(t+1) \in [-1, 0]$. Note that for $\boldsymbol{y}(t) \in [-1, 0]$, it holds that

$$\boldsymbol{y}(t)^{K-2}(\boldsymbol{y}(t)^K - 1) \leq 2.$$

This is due to the fact that for function $g(x) = x^{K-2}(x^K - 1)$ where $K > 1$ is odd, $g(x)$ is non-increasing in $[-1, 0]$ and therefore $g(x) \leq g(-1) = 2$. Due to the assumption that $\eta' < 1/4$, it holds that

$$\boldsymbol{y}(t+1) = \boldsymbol{y}(t)[1 - 2\eta'\boldsymbol{y}^{K-2}(t)(\boldsymbol{y}^K(t) - 1)] \leq 0.$$

Besides, since $\boldsymbol{y}^{K-1}(t)(\boldsymbol{y}^K(t) - 1) < 0$ always holds, it holds that $\boldsymbol{y}(t+1) \geq \boldsymbol{y}(t) \geq \boldsymbol{y}(0) = -1$. In summary, it holds that $\boldsymbol{y}(t) \in [-1, 0]$ for all $T$, leading to the conclusion that $|\boldsymbol{y}(t) - 1| \geq 1$.

**Adaptive learning rate.** Recall the iteration

$$\boldsymbol{y}(t+1) = \boldsymbol{y}(t) - 2\eta(\boldsymbol{y}^K(t) - 1). \tag{12}$$

This is equivalent to

$$1 - \boldsymbol{y}(t+1) = (1 - \boldsymbol{y}(t))(1 - 2\eta\frac{\boldsymbol{y}(t)^K - 1}{\boldsymbol{y}(t) - 1}).$$

Firstly, notice that $\frac{\boldsymbol{y}(t)^K - 1}{\boldsymbol{y}(t) - 1} \leq K$ when $\boldsymbol{y}(t) \in [-1, 1]$. Therefore, when $\eta < \frac{1}{2K}$ and $\boldsymbol{y}(0) = -1$, it holds that $\boldsymbol{y}(t) \leq 1$ for all $t$. Furthermore, since $\boldsymbol{y}^K(t) - 1 < 0$ always holds, $\boldsymbol{y}(t+1) \geq \boldsymbol{y}(t) \geq \boldsymbol{y}(0) = -1$. In summary, it holds that for all $t$,

$$\boldsymbol{y}(t) \in [-1, 1].$$

Besides, due to Fact C.2, it holds that

$$\frac{\boldsymbol{y}(t)^K - 1}{\boldsymbol{y}(t) - 1} \geq \frac{1}{K - 1}.$$

Therefore,

$$|1 - \boldsymbol{y}(t)| \leq (1 - \frac{2\eta}{K - 1})|1 - \boldsymbol{y}(t-1)| \leq 2(1 - \frac{2\eta}{K - 1})^T.$$

**Fact C.2.** For $x \in [-1, 1]$ and an odd $K > 1$, it holds that $\frac{x^K - 1}{x - 1} \geq \frac{1}{K-1}$.

*Proof.* Note that firstly, since $x^2 + x^3 > 0, \cdots, x^{K-3} + x^{K-2} > 0$,

$$\frac{x^K - 1}{x - 1} = 1 + x + x^2 + \cdots + x^{K-1} \geq 1 + x + x^{K-1}.$$

For function $g(x) = 1 + x + x^{K-1}$, it attains minimum at $x_0$ where $x_0^{K-2} = -\frac{1}{K-1}$. Therefore,

$$g(x) \geq g(x_0) = 1 + x_0 + x_0^{K-1} = 1 + \frac{K-2}{K-1}x_0 \geq \frac{1}{K-1}.$$

$\square$

### C.4. Proof of Theorem 3.3

Denote $R'(\boldsymbol{y}) = \frac{1}{K-M+1}\boldsymbol{y}^{K-M+1} + \epsilon\frac{1}{2-M}\boldsymbol{y}^{2-M}$.
Construct $\phi(\boldsymbol{y}) = f(\boldsymbol{y}^K(t)) + \lambda R'(\boldsymbol{y}(t))$, then

$$\phi(\boldsymbol{y}(t+1)) - \phi(\boldsymbol{y}(t))$$
$$= [\nabla\phi(\boldsymbol{y}(t))]^\top(\boldsymbol{y}(t+1) - \boldsymbol{y}(t)) + (\boldsymbol{y}(t+1) - \boldsymbol{y}(t))^\top \nabla^2\phi(\xi)(\boldsymbol{y}(t+1) - \boldsymbol{y}(t))$$

Note that

$$\boldsymbol{y}(t+1) - \boldsymbol{y}(t) = -\lambda\eta_t\boldsymbol{y}(t) - \eta_t\frac{\boldsymbol{y}^M(t)}{\boldsymbol{y}^{K-1}(t) + \epsilon\mathbf{1}} \odot \boldsymbol{y}^{K-1}(t) \odot \nabla f(\boldsymbol{y}^K(t))$$

$$= -\eta_t\frac{\boldsymbol{y}^M(t)}{\boldsymbol{y}^{K-1}(t) + \epsilon\mathbf{1}} \odot \boldsymbol{y}^{K-1}(t) \odot [\nabla f(\boldsymbol{y}^K(t)) + \lambda\boldsymbol{y}^{1-M}(t) + \epsilon\lambda\boldsymbol{y}^{2-M-K}(t)].$$

$$\nabla\phi(\boldsymbol{y}(t)) = \boldsymbol{y}^{K-1}(t)\nabla f(\boldsymbol{y}^K(t)) + \lambda\boldsymbol{y}^{K-M}(t) + \lambda\epsilon\boldsymbol{y}^{1-M}(t)$$
$$= \boldsymbol{y}^{K-1}(t) \odot [\nabla f(\boldsymbol{y}^K(t)) + \lambda\boldsymbol{y}^{1-M}(t) + \lambda\epsilon\boldsymbol{y}^{2-M-K}(t)].$$

Besides, when $\boldsymbol{y}(t)$ is bounded and $f$ is smooth, the function $\phi$ is also smooth. Combining the above discussion leads to

$$\phi(\boldsymbol{y}(t+1)) - \phi(\boldsymbol{y}(t))$$
$$= -\eta_t\|V'(\boldsymbol{y}(t)) \odot [\nabla f(\boldsymbol{y}^K(t)) + \lambda\boldsymbol{y}^{1-M}(t) + \lambda\epsilon\boldsymbol{y}^{2-M-K}(t)]\|^2 + \mathcal{O}(\eta_t^2),$$

where $V'(\boldsymbol{y}(t)) = \frac{\boldsymbol{y}^{M/2}(t)}{\sqrt{\boldsymbol{y}^{K-1}(t) + \epsilon\mathbf{1}}} \odot \boldsymbol{y}^{K-1}(t)$. By telescoping, it holds that

$$\sum_{t \in [T]} \eta_t\|V'(\boldsymbol{y}(t)) \odot [\nabla f(\boldsymbol{y}^K(t)) + \lambda\boldsymbol{y}^{1-M}(t) + \lambda\epsilon\boldsymbol{y}^{2-M-K}(t)]\|^2 \leq \phi(\boldsymbol{y}(0)) - \phi(\boldsymbol{y}(T)) + O(\sum_{t \in [T]} \eta_t^2).$$

By setting $\eta_t = 1/\sqrt{t}$, it holds that

$$\min_{t \in [T]} \|V'(\boldsymbol{y}(t)) \odot [\nabla f(\boldsymbol{y}^K(t)) + \lambda\boldsymbol{y}^{1-M}(t) + \lambda\epsilon\boldsymbol{y}^{2-M-K}(t)]\|^2 \leq \frac{\phi(\boldsymbol{y}(0)) - \phi(\boldsymbol{y}(T))}{\sqrt{T}} + \frac{\log T}{\sqrt{T}}.$$

Since $f(\cdot)$ and $\boldsymbol{y}(t)$ are bounded, it holds that $\phi(\boldsymbol{y}(t))$ is bounded for all $t \in [T]$. By rewriting $\boldsymbol{y}(t) = [\boldsymbol{x}(t)]^{1/K}$, it holds that

$$\min_{t \in [T]} \|V(\boldsymbol{x}(t)) \odot [\nabla f(\boldsymbol{x}(t)) + \lambda\boldsymbol{x}^{(1-M)/K}(t) + \lambda\epsilon\boldsymbol{x}^{(2-M-K)/K}(t)]\|^2 \leq \frac{\log T}{\sqrt{T}}.$$

Namely,

$$\min_{t \in [T]} \|V(\boldsymbol{x}(t)) \odot \nabla[f(\boldsymbol{x}(t)) + R(\boldsymbol{x})]\|^2 \leq \frac{\log T}{\sqrt{T}},$$

where $V(\boldsymbol{x}(t)) = \frac{\boldsymbol{x}^{M/(2K)}(t)}{\sqrt{\boldsymbol{x}^{1-1/K}(t) + \epsilon\mathbf{1}}} \odot \boldsymbol{x}^{1-1/K}(t)$.

*Remark* C.3 (On Theorem 3.3). Three aspects of Theorem 3.3 merit clarification. **(i) On characterizing $R(x)$ via stationarity.** Identifying the implicit regularizer through a stationarity-type condition follows the standard methodology in the implicit-bias literature (Gunasekar et al., 2018; Woodworth et al., 2020), where regularizers are characterized by matching limit points of the optimization trajectory to the KKT / stationarity conditions of a regularized objective. Theorem 3.3 follows this paradigm. **(ii) On the even-integer restriction on $M$.** The restriction to even integer $M$ is imposed purely for notational simplicity, consistent with prior reparameterization work such as PowerPropagation (Schwarz et al., 2021) and DWF (Kolb et al., 2023). For any real $M > 0$, one can equivalently replace $y^M$ with $|y|^M$ in the update to preserve its sign, and the conclusion of Theorem 3.3 continues to hold. **(iii) On the role of assumptions.** The formulations [B1], [Bp], and [Cp] are general problem definitions and intentionally impose no assumptions on $f$. Assumptions on $f$ (such as bounded smoothness here, or the Expansion Property in Assumption 3.13) are introduced only for specific theoretical results as needed, following standard practice in nonconvex optimization (Gunasekar et al., 2018; Kolb et al., 2023).

### C.5. Proof of Theorem 3.7

$\ell_1$ **clipping.** The Gradient Bound of $\ell_1$ approximation at point $u$ is its gradient on $u$,

$$\mathcal{E}_1 = \sqrt{d}pu^{p-1},$$

and the Approximation Error between $\ell_p$ norm and its $\ell_1$ approximation is

$$\mathcal{E}_2 = d(1-p)u^p, \tag{13}$$

which is the approximation error at the zero point. We note that $\psi(\mathbf{0}) = 0$ and $\tilde{\psi}(\mathbf{0}) = d(1-p)u^p$. By constraining $\mathcal{E}_1 \leq \sqrt{d}$, the clipping point satisfies

$$u \geq p^{1/(1-p)}.$$

In such a case, the approximation error would be

$$\mathcal{E}_2 = d(1-p)u^p \geq d(1-p)p^{p/(1-p)}.$$

Consider the function $F(p) = d(1-p)p^{p/(1-p)}$ with gradient $\frac{p^{p/(1-p)} \log p}{1-p} < 0$, therefore, the clipping error would be larger than the value at $p = 1 - \frac{1}{cd}$, therefore,

$$\mathcal{E}_2 > \frac{1}{c}\left(1 - \frac{1}{cd}\right)^{cd-1} > \frac{1}{ce}.$$

**Constant clipping.** The Gradient Bound of $\ell_1$ approximation at point $u$ is its gradient on $u$

$$\mathcal{E}_1 = \sqrt{d}pu^{p-1},$$

and the Approximation Error between $\ell_p$ norm and its $\ell_1$ approximation is

$$\mathcal{E}_2 = du^p, \tag{14}$$

which is the approximation error at the zero point. We note that $\psi(\mathbf{0}) = 0$ and $\tilde{\psi}(\mathbf{0}) = du^p$. By constraining $\mathcal{E}_1 \leq \sqrt{d}$, it holds that

$$u \geq p^{1/(1-p)}.$$

In such a case, the approximation error would be

$$\mathcal{E}_2 = du^p \geq dp^{p/(1-p)}.$$

Consider the function $F(p) = dp^{p/(1-p)}$ with gradient $\frac{p^{p/(1-p)}(1-p+\log p)}{(1-p)^2} < 0$, therefore, the clipping error would be larger than the value at $p = 1$, therefore,

$$\mathcal{E}_2 \geq \lim_{p \to 1} dp^{p/(1-p)} \geq d/e.$$

### C.6. Proof of Example 3.8

Due to the symmetry of the initialization and the update rule, we observe that $\boldsymbol{y}_1(T) = \boldsymbol{y}_2(T) = \cdots = \boldsymbol{y}_K(T)$ for all $T > 0$. We simplify the notation by denoting the common parameter vector as $\boldsymbol{y}(T)$.

**(a.) Without Weight Decay** We first analyze the iteration based on the unpenalized loss $\tilde{L}(\boldsymbol{y})$ using gradient descent with a constant learning rate $\eta$:

$$\tilde{\boldsymbol{y}}(T+1) = \tilde{\boldsymbol{y}}(T) - \eta\tilde{\boldsymbol{y}}(T)^{K-1} \odot \nabla f(\tilde{\boldsymbol{x}}(T)).$$

Note that the gradient can be expressed as $\nabla f(\boldsymbol{x}) = \Lambda\boldsymbol{x}$. The update then becomes:

$$\tilde{\boldsymbol{y}}(T+1) = \tilde{\boldsymbol{y}}(T) - \eta\Lambda\tilde{\boldsymbol{y}}(T)^{(2K-1)}.$$

Let $\lambda_i$ be the $i$-th eigenvalue of $\Lambda$. Without loss of generality, assume $\lambda_1 \geq \cdots \geq \lambda_R > \lambda_{R+1} = \cdots = \lambda_d = 0$. For any coordinate $i \in \{R+1, \ldots, d\}$, it holds that:

$$\tilde{\boldsymbol{y}}(T+1)[i] = \tilde{\boldsymbol{y}}(T)[i] = \cdots = \tilde{\boldsymbol{y}}(0)[i] = \alpha.$$

Since $\alpha = \Theta(1)$, the $i$-th coordinate of the reparameterized input $\tilde{\boldsymbol{x}}$ remains non-zero for all $T$. Consequently:

$$\| \odot_{k\in[K]} \tilde{\boldsymbol{y}}_k(T)\|_0 \geq d - R.$$

**(b.) With Weight Decay** We next consider the penalized loss $L(\boldsymbol{y}) = f(\odot\boldsymbol{y}_k) + \lambda\sum\|\boldsymbol{y}_k\|_2^2$. At convergence to a stationary point $\boldsymbol{y}(\infty)$, the gradient must vanish:

$$\nabla L(\boldsymbol{y}(\infty)) = \boldsymbol{y}(\infty)^{K-1} \odot \Lambda\boldsymbol{y}(\infty)^K + 2\lambda K\boldsymbol{y}(\infty) = \boldsymbol{0}.$$

Analyzing the $i$-th coordinate:

- **Case 1: $\Lambda_i = 0$.** The equation reduces to $2\lambda K\boldsymbol{y}(\infty)[i] = 0$. Since $\lambda, K > 0$, it follows that $\boldsymbol{y}(\infty)[i] = 0$.

- **Case 2: $\Lambda_i \neq 0$.** The condition is $\boldsymbol{y}(\infty)[i] \left(\Lambda_i\boldsymbol{y}(\infty)[i]^{2K-2} + 2\lambda K\right) = 0$. This implies either $\boldsymbol{y}(\infty)[i] = 0$ or $\boldsymbol{y}(\infty)[i]^{2K-2} = -\frac{2\lambda K}{\Lambda_i}$.

In both cases, as $\lambda \to 0$, we have $\boldsymbol{y}(\infty)[i] \to 0$. Thus, the $\ell_0$ norm of the resulting reparameterized vector vanishes in the limit:

$$\| \lim_{\lambda\to 0} \odot_{k\in[K]}\boldsymbol{y}_k(\infty)\|_0 = 0.$$

### C.7. Proof of Theorem 3.9

The proof of Theorem 3.9 performs similarly to that of Example 3.8. Due to the symmetry of the initialization and the update rule, we observe that $\boldsymbol{y}_1(T) = \boldsymbol{y}_2(T) = \cdots = \boldsymbol{y}_K(T)$ for all $T > 0$. We simplify the notation by denoting the common parameter vector as $\boldsymbol{y}(T)$.

**(a.) Without Weight Decay** Consider the gradient descent trajectory for the unpenalized objective. The parameters at time $T+1$ can be expressed as:

$$\tilde{\boldsymbol{y}}(T+1) = \tilde{\boldsymbol{y}}(0) - \eta\sum_{t=0}^{T}\Delta_t.$$

By assumption, the update term $\Delta_t$ is always confined to the fixed subspace $\mathcal{S}$ of dimension $R$. Let $\mathcal{S}^\perp$ be the orthogonal complement (the kernel space) of $\mathcal{S}$ with dimension $d - R$. For any vector $\mathbf{v} \in \mathcal{S}^\perp$, it holds that $\mathbf{v}^\top\Delta_t = 0$. Thus:

$$\mathbf{v}^\top\tilde{\boldsymbol{y}}(T+1) = \mathbf{v}^\top\tilde{\boldsymbol{y}}(0).$$

Since the initialization $\tilde{\boldsymbol{y}}(0) = \Theta(1)$ is non-zero, the projection of the parameter vector onto $\mathcal{S}^\perp$ remains constant and large for all $T$. Consequently, at least $d - R$ coordinates of the resulting product $\boldsymbol{x}$ cannot reach zero, implying $\| \odot_{k\in[K]} \tilde{\boldsymbol{y}}_k(T)\|_0 \geq d - R$.

**(b.) With Weight Decay** Now consider the penalized loss $\mathcal{L}(\boldsymbol{y}) = f(\bigodot \boldsymbol{y}_k) + \lambda \sum \|\boldsymbol{y}_k\|_2^2$. The weight decay term $\lambda\|\boldsymbol{y}\|_2^2$ is isotropic; its gradient $2\lambda\boldsymbol{y}$ acts on all coordinates, including those in $\mathcal{S}^\perp$. A stationary point $\boldsymbol{y}(\infty)$ must satisfy:

$$\nabla_{\boldsymbol{y}} f(\boldsymbol{x}) + 2K\lambda\boldsymbol{y} = \boldsymbol{0}.$$

By the chain rule, $\nabla_{\boldsymbol{y}} f(\boldsymbol{x}) = \nabla_{\boldsymbol{x}} f(\boldsymbol{x}) \odot \frac{\partial \boldsymbol{x}}{\partial \boldsymbol{y}}$. Near the origin, the first-order behavior is governed by the Hessian $\nabla^2 f(\boldsymbol{0})$. The condition $\lambda \geq \frac{1}{2K}(-\sigma_{\min}(\nabla^2 f(\boldsymbol{x})))$ ensures that the pull of the weight decay toward the origin is stronger than any local "push" from the objective function's curvature. In this regime, the origin becomes the only stable stationary point. Therefore, the parameters converge to zero, yielding $\|\lim_{\lambda \to 0} \bigodot_{k \in [K]} \boldsymbol{y}_k(\infty)\|_0 = 0$.

*Remark* C.4 (Mildness of the Low-Dimensional Update Subspace Condition). The Low-Dimensional Update Subspace condition is mild and is satisfied in several standard settings. (1) **NTK regime** (Jacot et al., 2018): NTK stays constant, so updates lie in a fixed subspace with dimension bounded by sample count. (2) **Lazy training** (Chizat et al., 2019): overparameterization confines updates to the tangent space at initialization. (3) **Sparse linear models**: the condition is automatically satisfied, which is covered by our experiments on synthetic dataset.

### C.8. Proof of Theorem 3.10

**Case 1: Adaptive Learning Rate with $M$** The iteration for ReWA with parameter $M$ is given by:

$$y(t+1) = (1 - \lambda\eta_t)y(t) - \eta_t y^M(t)\nabla f(y^K(t))$$

To remain in the stagnation region, the magnitude of the update must be less than the magnitude of the current state:

$$|\eta_t y^M(t)\nabla f(y^K(t))| < |(1 - \lambda\eta_t)y(t)|$$

Using the bound $|\nabla f| \leq B$ and assuming $1 - \lambda\eta_t > 0$:

$$\eta_t |y(t)|^M B < (1 - \lambda\eta_t)|y(t)| \tag{15}$$

$$|y(t)|^{M-1} < \frac{1 - \lambda\eta_t}{\eta_t B} \tag{16}$$

Thus, the stagnation region is defined by:

$$|y(t)| < \left[\frac{1 - \lambda\eta_t}{\eta_t B}\right]^{1/(M-1)} \triangleq U_1$$

**Case 2: Adaptive Learning Rate with $\epsilon$** The iteration with smoothing parameter $\epsilon$ is:

$$y(t+1) = (1 - \lambda\eta_t)y(t) - \eta_t \frac{y^{K-1}(t)}{y^{K-1}(t) + \epsilon}\nabla f(y^K(t))$$

The condition $y(t)y(t+1) > 0$ requires:

$$\eta_t \frac{|y(t)|^{K-1}}{|y(t)|^{K-1} + \epsilon}B < (1 - \lambda\eta_t)|y(t)|$$

Dividing by $|y(t)|$:

$$\frac{|y(t)|^{K-2}}{|y(t)|^{K-1} + \epsilon} < \frac{1 - \lambda\eta_t}{\eta_t B}$$

That is to say:

$$\frac{|y(t)|^{K-1} + \epsilon}{|y(t)|^{K-2}} > \frac{\eta_t B}{1 - \lambda\eta_t} \tag{17}$$

$$|y(t)| + \epsilon|y(t)|^{-(K-2)} > \frac{\eta_t B}{1 - \lambda\eta_t} \tag{18}$$

Let $F(U) = U + \epsilon U^{-(K-2)}$. The boundary of this region is defined by the value $U_2$ such that $F(U_2) = \frac{\eta_t B}{1 - \lambda\eta_t}$, hence:

$$|y(t)| \leq U_2 \triangleq F^{-1}\left(\frac{\eta_t B}{1 - \lambda\eta_t}\right)$$

**Case 3: Without Adaptive Learning Rate**    When no adaptive learning rate is applied, the update is:

$$y(t+1) = (1 - \lambda\eta_t)y(t) - \eta_t y^{K-1}(t)\nabla f(y^K(t))$$

Following the same logic as Case 1:

$$\eta_t|y(t)|^{K-1}B < (1 - \lambda\eta_t)|y(t)| \tag{19}$$

$$|y(t)|^{K-2} < \frac{1 - \lambda\eta_t}{\eta_t B} \tag{20}$$

The stagnation region is:

$$|y(t)| < \left[\frac{1 - \lambda\eta_t}{\eta_t B}\right]^{1/(K-2)} \triangleq U_3$$

### C.9. Proof of Proposition 3.11

Proposition 2.11 compares the sizes of these stagnation regions to demonstrate the benefits of the adaptive learning rate. Let $C = \frac{1-\lambda\eta_t}{\eta_t B}$.

**Comparison of $U_1$ and $U_3$**    Recall $U_1 = C^{1/(M-1)}$ and $U_3 = C^{1/(K-2)}$. We consider the case where the optimization is refined (i.e., $C \leq 1$). Since $M < K - 1$, it follows that:

$$M - 1 < K - 2 \implies \frac{1}{M-1} > \frac{1}{K-2}$$

For any $0 < C \leq 1$, a larger exponent results in a smaller value. Therefore:

$$C^{1/(M-1)} \leq C^{1/(K-2)} \implies U_1 \leq U_3$$

This proves that the adaptive parameter $M$ reduces the region where the sign of $y(t)$ cannot be changed.

**Comparison of $U_2$ and $U_3$**    To compare $U_2$ and $U_3$, we analyze the function $F(U) = U + \epsilon U^{-(K-2)}$ for $U > 0$.

Firstly, note that the derivative is $F'(U) = 1 - \epsilon(K-2)U^{-(K-1)}$. $F(U)$ is strictly decreasing when $F'(U) < 0$, which occurs for:

$$U < [\epsilon(K-2)]^{1/(K-1)}$$

In the context of stagnation near the origin, we are concerned with the branch of $F^{-1}$ that approaches $0$ as the threshold $1/C$ approaches infinity. On this branch, $F(U)$ is strictly decreasing.

From the definition of $U_3$, we have $U_3^{K-2} = C$, or $U_3^{-(K-2)} = 1/C$. Substituting $U_3$ into $F(U)$:

$$F(U_3) = U_3 + \epsilon U_3^{-(K-2)} = C^{1/(K-2)} + \frac{\epsilon}{C}$$

Since $F(U)$ is decreasing in the region of interest, the inequality $U_2 \leq U_3$ is equivalent to $F(U_2) \geq F(U_3)$. Given $F(U_2) = 1/C$, we require:

$$\frac{1}{C} \geq C^{1/(K-2)} + \frac{\epsilon}{C} \implies \frac{1 - \epsilon}{C} \geq C^{1/(K-2)}$$

Rearranging this inequality:

$$1 - \epsilon \geq C \cdot C^{1/(K-2)} = C^{(K-1)/(K-2)} \tag{21}$$

$$C \leq (1 - \epsilon)^{(K-2)/(K-1)} \tag{22}$$

This confirms that under the specified bound on $C$, the stagnation radius with adaptive $\epsilon$ is smaller than or equal to the non-adaptive radius ($U_2 \leq U_3$).

**Divergence Case** If $C > 1$, then $U_1, U_2, U_3$ all become greater than 1. This implies that the stagnation region covers the entire unit ball around the origin, indicating that the learning rate $\eta_t$ is too small relative to the gradient bound $B$ to cross the origin regardless of the adaptive scheme.

### C.10. Proof of Example 3.12

We consider the constraint $\|\theta - \theta_u\|_2^2 \leq (d-1)u^2$, where $\theta_u = (u, \cdots, u)$ and $u > 0$.

$\ell_0$**-optimization.** The optimal vector for $\ell_0$-optimization problem satisfies $\|\theta_0^*\| = 1$.

Firstly, notice that if $\|\theta_0^*\| = 0$, $\|\theta - \theta_u\|_2^2 = du^2 > (d-1)u^2$ which violets the constraints. Therefore, $\|\theta_0^*\|_0 \geq 1$.

Secondly, notice that the vector $\theta_0 = (u, 0, \cdots, 0)$ satisfies both $\|\theta_0^*\|_0 = 1$ and $\|\theta - \theta_u\|_2^2 \leq (d-1)u^2$.

$\ell_1$**-optimization.** The optimal vector for $\ell_0$-optimization problem satisfies $\|\theta_1^*\| = d$.

Consider the hyperplane $x_1 + x_2 + ... + x_d = d(1 - \sqrt{\frac{d-1}{d}})u$.

Firstly, note that the $\ell_1$ ball with radius $d(1 - \sqrt{\frac{d-1}{d}})u$ all satisfies $x_1 + x_2 + ... + x_d \leq d(1 - \sqrt{\frac{d-1}{d}})u$.

Secondly, note that the ball satisfies $x_1 + x_2 + ... + x_d \leq d(1 - \sqrt{\frac{d-1}{d}})u$ because the distance between the center and the hyperplane is lager than the radius.

Therefore, the hyperplane ensures that $\theta_1^* = ((1 - \sqrt{\frac{d-1}{d}})u, \cdots, (1 - \sqrt{\frac{d-1}{d}})u)$, leading to the fact that $\|\theta_1^*\|_0 = d$.

$\ell_p$**-optimization.** The optimal vector for $\ell_p$-optimization problem satisfies $\|\theta_p^*\| = 1$, for $p \leq 1/2$.

Assume that the optimal vector is $\theta_p^* = (m_1, \cdots, m_n)$. Our goal is to show that $\|\theta_p^*\|_0 = 1$.

*Claim 1: The high dimensional case is equivalent to a low-dimensional case.*

We use the greedy algorithm to reduce the problem. We will show that for any two dimensions (e.g., $m_1, m_2$, WLOG, assume that $m_1 \geq m_2 > 0$), it is always a better choice to put all mass on only one dimension. Therefore, greedy doing so leads to the conclusion that $\|\theta_p^*\|_0 = 1$.

We next show the greedy step holds. Formally, it suffices to show that for any $\bar{m}_1 > m_1 \geq m_2 > \bar{m}_2 = 0$, if $(\bar{m}_1 - u)^2 + u^2 = (m_1 - u)^2 + (m_2 - u)^2$, we have

$$\bar{m}_1^p < m_1^p + m_2^p.$$

One may wonder whether there always exists $\bar{m}_2 = 0$. It turns out that we can wlog assume that $(m_1 - 1)^2 + (m_2 - 1)^2 > 1$ (otherwise the solution is not in the boundary)

We plot a fig in 2-dim, it suffices to show that the circle only intersect with the $\ell_p$ ball in the endpoint. The next goal is to prove that the ball $(x - u)^2 + (y - u)^2 \leq (\bar{m}_1 - u)^2 + u^2$ and the $\ell_p$-ball $x^p + y^p \leq \bar{m}_1^p$ only intersect on the endpoint.

We first use the polar coordinates, which leads to

$$\rho_1 = \frac{\bar{m}_1}{(\cos^p \theta + \sin^p \theta)^{1/p}},$$
$$\rho_2 = u(\sin \theta + \cos \theta) + \sqrt{(\bar{m}_1 - u)^2 - u^2 + u^2(\sin \theta + \cos \theta)^2}.$$

It suffices to prove that $\rho_1 < \rho_2$.

$$\rho_1 < \rho_2$$

$$\Longleftrightarrow \frac{\bar{m}_1}{(\cos^p \theta + \sin^p \theta)^{1/p}} < u(\sin \theta + \cos \theta) + \sqrt{(\bar{m}_1 - u)^2 - u^2 + u^2(\sin \theta + \cos \theta)^2}$$

$$\Longleftrightarrow \frac{b}{(\cos^p \theta + \sin^p \theta)^{1/p}} < (\sin \theta + \cos \theta) + \sqrt{(b-1)^2 - 1 + (\sin \theta + \cos \theta)^2}$$

$$\Longleftrightarrow (b-1)^2 - 1 + (\sin \theta + \cos \theta)^2 < [(\sin \theta + \cos \theta) - \frac{b}{(\cos^p \theta + \sin^p \theta)^{1/p}}]^2$$

$$\Longleftrightarrow (b-2)(\cos^p \theta + \sin^p \theta)^{2/p} < b - 2(\cos \theta + \sin \theta)(\cos^p \theta + \sin^p \theta)^{1/p}$$

We scale with u in the second line and denote $b = \bar{m}_1 / u \in (0, 1)$.

To let the final line holds, notice that it is linear in $b$. We only requires that is holds when $b = 0$ and $b = 1$. When $b = 0$, it naturally hold when $p < 1$ that

$$-2(\cos^p \theta + \sin^p \theta)^{1/p} < -2(\cos \theta + \sin \theta).$$

When $b = 1$, we require that

$$\Longleftrightarrow [(\cos^p \theta + \sin^p \theta)^{1/p} - (\sin \theta + \cos \theta)]^2 \geq (\sin \theta + \cos \theta)^2 - 1.$$

Notice that $(\cos^p \theta + \sin^p \theta)^{1/p} - (\sin \theta + \cos \theta) > 0$, it is equivalent to

$$(\cos^p \theta + \sin^p \theta)^{1/p} \geq (\sin \theta + \cos \theta) + \sqrt{(\sin \theta + \cos \theta)^2 - 1}$$
$$= \sin \theta + \cos \theta + \sqrt{2}\sqrt{\sin \theta \cos \theta}$$
$$\leq [\sqrt{\sin \theta} + \sqrt{\cos \theta}]^2,$$

which holds when $p < 0.5$.

### C.11. Proof of Theorem 3.14

In this section, we prove that the global minimizer of [Cp] is approximately sparse. The proof outline are as follows. Firstly, Lemma 3.1 guarantees that the global minimizer of [Cp] is just the same as that of [Bp]. Secondly, Lemma C.5 shows that there exists a penalty $\lambda$ such that the global minimizer of [Bp] is just the solution of [Ap]. Finally, Lemma C.6 demonstrates that the solution of [Ap] is approximately sparse, that is to say, by eliminating the small atoms, its zero norm can be bounded with that of [A0].

**Lemma C.5** (Relationship between [Bp] and [Ap]). *Assume that $f(\boldsymbol{x})$ satisfies Assumption 3.13, and assume that the solution of [Bp] and its corresponding expansion point has bounded p-norm. Then as $\lambda \to 0$, the global minimizer of [Bp], if exists, converges to a solution of [Ap].*

*Proof of Lemma C.5.* Recall the two problems of [Bp] and [Ap]. Denote $\boldsymbol{x}^*$ as the minimizer of [Bp] and $\boldsymbol{x}_p$ as the minimizer of [Ap]. Besides, denote $\hat{\boldsymbol{x}}^*$ as the expansion point to $\boldsymbol{x}^*$ in Assumption 3.13. Denote $B_0$ as the bound of the solution of [Bp] and its corresponding expansion point.

Due to the optimality of [Bp] and [Ap], it holds that

1. $f(\boldsymbol{x}^*) + \lambda \|\boldsymbol{x}^*\|_p^p \leq f(\boldsymbol{x}_p) + \lambda \|\boldsymbol{x}_p\|_p^p$,

2. $f(\boldsymbol{x}^*) + \lambda \|\boldsymbol{x}^*\|_p^p \leq f(\hat{\boldsymbol{x}}^*) + \lambda \|\hat{\boldsymbol{x}}^*\|_p^p$,

3. $f(\boldsymbol{x}_p) = f(\hat{\boldsymbol{x}}^*) \leq f(\boldsymbol{x}^*)$,

4. $\|\boldsymbol{x}^*\|_p \leq \|\boldsymbol{x}_p\|_p \leq \|\hat{\boldsymbol{x}}^*\|_p$,

where the last equation is from a simple calculation of the first three arguments.

We next prove by contradiction that $\|\hat{\boldsymbol{x}}^*\|_p^p \leq \|\boldsymbol{x}^*\|_p^p$. To do so, we assume that $\|\hat{\boldsymbol{x}}^*\|_p^p \geq (1+\delta)\|\boldsymbol{x}^*\|_p^p$ for any small constant $\delta > 0$. Under this condition,

$$\mu M_{\delta,\beta}\|\hat{\boldsymbol{x}}^*\|_p^\beta - \mu M_{\delta,\beta}\|\boldsymbol{x}^*\|_p^\beta \leq \mu\|\hat{\boldsymbol{x}}^* - \boldsymbol{x}^*\|_p^\beta \leq f(\boldsymbol{x}^*) - f(\hat{\boldsymbol{x}}^*) \leq \lambda\|\hat{\boldsymbol{x}}^*\|_p^p - \lambda\|\boldsymbol{x}^*\|_p^p,$$

where $M_{\delta,\beta} = \frac{\delta^{\beta/p}}{(1+\delta)^{\beta/p}-1}$. Notably, the first inequality comes from Lemma C.7, the second comes from the definition of expansion and the last inequality comes from the second argument above.

Therefore, it holds that

$$\mu M_{\delta,\beta}\|\hat{\boldsymbol{x}}^*\|_p - \lambda\|\hat{\boldsymbol{x}}^*\|_p^p \leq \mu M_{\delta,\beta}\|\boldsymbol{x}^*\|_p - \lambda\|\boldsymbol{x}^*\|_p^p.$$

Consider the function $\psi(u) = \mu M_{\delta,\beta}u - \lambda u^p$, which is increasing in $(0, B_0)$ when $\lambda < \frac{\mu M_{\delta,\beta}}{pB_0^{p-1}}$. Note that we consider a case $\lambda \to 0$. Therefore, for any given fixed $\delta > 0$, there would exist a sufficiently small $\lambda$ such that $\lambda < \frac{\mu M_{\delta,\beta}}{pB_0^{p-1}}$. In such a case, $\|\hat{\boldsymbol{x}}^*\|_p \leq \|\boldsymbol{x}^*\|_p$, according to the non-decreasing property. This contradicts with the assumption that $\|\hat{\boldsymbol{x}}^*\|_p^p \geq (1+\delta)\|\boldsymbol{x}^*\|_p^p$. Besides, note that $\delta$ can be arbitrarily small. Therefore, $\|\hat{\boldsymbol{x}}^*\|_p^p \leq \|\boldsymbol{x}^*\|_p^p$.

Combining the above discussion with the fourth argument, it holds that

$$\begin{aligned}\|\hat{\boldsymbol{x}}^*\|_p &= \|\boldsymbol{x}^*\|_p = \|\boldsymbol{x}_p\|_p, \\ f(\hat{\boldsymbol{x}}^*) &= f(\boldsymbol{x}^*) = f(\boldsymbol{x}_p),\end{aligned} \tag{23}$$

which means that $\boldsymbol{x}^*$ is also a solution of [Ap].

$\square$

**Lemma C.6** (Relationship between [Ap] and [A0]). *Let $\boldsymbol{x}_0$ denote the solution of [A0] with $\|\boldsymbol{x}_0\|_\infty \leq B$, and $\boldsymbol{x}_p$ denote the solution of [Ap] for a given $p \in (0,1)$. For a given constant $\gamma > 0$, $\bar{\boldsymbol{x}}_p \triangleq \max\{\boldsymbol{x}_p, \exp(-\gamma/\sqrt{p})\}$ eliminate small values of $\boldsymbol{x}_p$. Then if $p \leq \min\{\frac{1}{\gamma^2}, \frac{2}{B'}, (\frac{4B'+\gamma}{4B'+2\gamma^2+4B'\gamma})^2\}$ where $B' = \max\{0, \log B\}$, it holds that*

$$\|\bar{\boldsymbol{x}}_p\|_0 \leq (1 + (4B' + \gamma)\sqrt{p})\|\boldsymbol{x}_0\|_0.$$

*Proof of Lemma C.6.* Let $\boldsymbol{x}_0$ denote the solution of [A0] and $\boldsymbol{x}_p$ denote the solution of [Ap]. Let $\bar{\boldsymbol{x}}_p \triangleq \max\{\boldsymbol{x}_p, \exp(-\gamma/\sqrt{p})\}$ denotes the truncated version of $\boldsymbol{x}_p$, where $\gamma$ denotes a constant. We denote $\cdot[i]$ as the $i$-th coordinate.

Due to the optimality of $\boldsymbol{x}_0$ and $x_p$, it holds that

1. $\|\boldsymbol{x}_0\|_0 \leq \|\boldsymbol{x}_p\|_0$,

2. $\|\boldsymbol{x}_p\|_p \leq \|\boldsymbol{x}_0\|_p$,

3. $f(\boldsymbol{x}_0) = f(\boldsymbol{x}_p)$.

Firstly, for the truncated solution $\bar{\boldsymbol{x}}_p$, one can bound its 0-norm using $p$-norm,

$$\|\bar{\boldsymbol{x}}_p\|_p^p = \sum_{i \in [d]}(\bar{\boldsymbol{x}}_p[i])^p = \sum_{i:\bar{\boldsymbol{x}}_p[i]\neq 0}\exp(p\log\bar{\boldsymbol{x}}_p[i]) \geq \sum_{i:\bar{\boldsymbol{x}}_p[i]\neq 0}1 + p\log\bar{\boldsymbol{x}}_p[i] \geq (1-\gamma\sqrt{p})\|\bar{\boldsymbol{x}}_p\|_0,$$

where the last equation is due to the truncation property of $\bar{\boldsymbol{x}}_p$. Similarly for $\boldsymbol{x}_0$, it holds that

$$\|\boldsymbol{x}_0\|_p^p = \sum_{i:\boldsymbol{x}_0[i]\neq 0}\exp(p\log\boldsymbol{x}_0[i]) \leq \sum_{i:\boldsymbol{x}_0[i]\neq 0}1 + 4pB' \leq (1+4pB')\|\boldsymbol{x}_0\|_0,$$

where $B' = \max\{0, \log\|\boldsymbol{x}_0\|_\infty\}$. Notably, the second equation is derived from Lemma C.10. Therefore, combining the above discussion, we derive that

$$\|\bar{\boldsymbol{x}}_p\|_0 \leq \frac{1}{1-\gamma\sqrt{p}}\|\bar{\boldsymbol{x}}_p\|_p^p \leq \frac{1}{1-\gamma\sqrt{p}}\|\boldsymbol{x}_p\|_p^p \leq \frac{1}{1-\gamma\sqrt{p}}\|\boldsymbol{x}_0\|_p^p \leq \frac{1+4pB'}{1-\gamma\sqrt{p}}\|\boldsymbol{x}_0\|_0 \leq (1+(4B'+2\gamma)\sqrt{p})\|\boldsymbol{x}_0\|_0,$$

where the second equation is due to the truncation property, and the last equation is because $p \leq (\frac{4B'+\gamma}{4B'+2\gamma^2+4B'\gamma})^2$ under Fact C.9.

$\square$

**Lemma C.7** (Triangle Inequality for $p$-norm). *If $\frac{\|\hat{\boldsymbol{x}}^*\|_p^p}{\|\boldsymbol{x}^*\|_p^p} \geq 1 + \delta$ and $0 < p < 1$,*

$$M_{\delta,\beta}\|\hat{\boldsymbol{x}}^*\|_p^\beta - \|\boldsymbol{x}_1^*\|_p^\beta \leq \|\hat{\boldsymbol{x}}^* - \boldsymbol{x}^*\|_p^\beta,$$

*where $M_{\delta,\beta} = \frac{\delta^{\beta/p}}{(1+\delta)^{\beta/p}-1}$.*

*Proof.* Note that due to the concavity of $\|\cdot\|_p^p$, it holds that

$$\|\hat{\boldsymbol{x}}^* - \boldsymbol{x}^*\|_p^p \geq \|\hat{\boldsymbol{x}}^*\|_p^p - \|\boldsymbol{x}^*\|_p^p.$$

By setting $a = \|\hat{\boldsymbol{x}}^*\|_p^p$, $b = \|\boldsymbol{x}_1^*\|_p^p$ and $c = \beta/p$ in Fact C.8, it holds that

$$(\|\hat{\boldsymbol{x}}^*\|_p^p - \|\boldsymbol{x}_1^*\|_p^p)^{\beta/p} \geq M_{\delta,\beta}(\|\hat{\boldsymbol{x}}^*\|_p^\beta - \|\boldsymbol{x}_1^*\|_p^\beta).$$

Therefore, combining the two inequalities together leads to

$$M_{\delta,\beta}\|\hat{\boldsymbol{x}}^*\|_p^\beta - \|\boldsymbol{x}_1^*\|_p^\beta \leq \|\hat{\boldsymbol{x}}^* - \boldsymbol{x}^*\|_p^\beta.$$

$\square$

**Fact C.8.** If $a/b > 1 + \delta$, it holds that for $c > 1$,

$$(a-b)^c \geq M_{\delta,\beta}(a^c - b^c),$$

where $M_{\delta,\beta} = \frac{\delta^c}{(1+\delta)^c-1}$ and $c = \beta/p$.

*Proof of Fact C.8.* We firstly note that function $F(t) = \frac{(t-1)^c}{t^c-1}$ is non-decreasing for $t > 1$ and $c > 1$. This can be derived from its derivation $\frac{d}{dt}F(t) = \frac{c(t-1)^{c-1}}{(t^c-1)^2}(t^{c-1}-1)$. Therefore, by plugging into $t = a/b > 1 + \delta$, it holds that

$$F(t) = \frac{(t-1)^c}{t^c-1} \geq F(1+\delta) = \frac{\delta^c}{(1+\delta)^c-1}.$$

Therefore, it holds that
$$(a-b)^c \geq M_{\delta,\beta}(a^c - b^c).$$

$\square$

**Fact C.9.** $\frac{1+ap}{1-b\sqrt{p}} \leq 1 + (a+2b)\sqrt{p}$ if $p \leq (\frac{a+b}{a+2b^2+ab})^2$.

*Proof of Fact C.9.* It suffices to show that when $p \leq (\frac{a+b}{a+2b^2+ab})^2$

$$1 + ap \leq (1 + (a+2b)\sqrt{p})(1 - b\sqrt{p}).$$

By simple calculation, it suffices to show that

$$(a + ab + 2b^2)\sqrt{p} \leq a + b.$$

$\square$

**Lemma C.10.** *If $p\max_i \log \boldsymbol{x}_0[i] \leq 2$, it holds that $\exp(p\log \boldsymbol{x}_0[i]) \leq 1 + 4pB'$, where $B' = \max\{0, \log B\}$.*

*Proof of Lemma C.10.* Consider the function $\exp(u)$, it holds that

1. If $u \in [0, 2]$, $\exp(u) \leq 1 + 4u$,

2. If $u \in (-\infty, 0)$, $\exp(u) \leq 1$.

Therefore, if $p \log \boldsymbol{x}_0[i] \leq 2$, it holds that

$$\exp(p \log \boldsymbol{x}_0[i]) \leq 1 + 4p \max\{0, \log \boldsymbol{x}_0[i]\} \leq 1 + 4pB'.$$

$\square$

# D. Experiment Details

## D.1. Synthetic Dataset

We set $\beta^*$ as a nearly all-zero 10,000-dimensional vector, except for the first dimension to be 1.

**Ablation study of $K$ and $M$ (Figure 1).** We perform a grid search over $K \in \{3, 5, 7, 9, 11, 13\}$ and $M \in \{0, 2, 4, 6, 8, 10\}$ subject to $M < K$, using the following fixed hyperparameters: dimension $d = 10,000$, dataset size 2,000, batch size 25, 800 epochs, learning rate $2 \times 10^{-4}$ with cosine schedule, and weight decay $\kappa = 5 \times 10^{-5}$, $\varepsilon = 0$.

**Hyperparameter sensitivity analysis (Figure 2).** All four subfigures share the same base setting: $d = 10,000$, dataset size 2,000, batch size 25, 800 epochs, learning rate $2 \times 10^{-4}$ with cosine schedule. Sparsity is measured at thresholds $\{10^{-7}, 5 \times 10^{-7}, 10^{-6}, 5 \times 10^{-6}, 10^{-5}, 5 \times 10^{-5}, 10^{-4}, 5 \times 10^{-4}, 10^{-3}\}$, and each configuration is run 3 times (mean $\pm$ std reported).

- **(a) Algorithms.** Compares ReWA ($K = 9$, $M = 5$, $\varepsilon = 0$, $\lambda = 5 \times 10^{-5}$), SGD-$\ell_1$ ($K = 1$, $M = 0$, $\varepsilon = 0$, $\lambda = 5 \times 10^{-5}$), and Lasso.

- **(b) Reparameterization $K$.** Fixes $\varepsilon = 0$, $\lambda = 5 \times 10^{-5}$, and varies $(K, M) \in \{(9, 5), (7, 4), (5, 2), (3, 0), (1, 0)\}$.

- **(c) Adaptive learning rate $M$.** Fixes $K = 9$, $\varepsilon = 0$, $\lambda = 5 \times 10^{-5}$, and varies $M \in \{0, 2, 5\}$.

- **(d) Weight decay $\lambda$.** Fixes $K = 9$, $M = 5$, $\varepsilon = 0$, and varies $\lambda \in \{10^{-5}, 2.5 \times 10^{-5}, 5 \times 10^{-5}, 10^{-4}\}$.

## D.2. CIFAR10

We directly follow the experimental setting of Spred (Liu & Wang, 2023). Specifically, we train a ResNet18 model (He et al., 2016) on the CIFAR10 (Krizhevsky et al., 2009) dataset. We use a learning rate initially set to 0.256, a weight decay of 1e-4, a momentum of 0.875, and a batch size of 256 for 100 epochs. During the first 5 epochs, we employ a linear warmup, and then apply a cosine scheduler to adjust the learning rate. Specially, for $\ell_1$ case, we set the $\ell_1$ weight decay to 1e-5.

During the evaluation, we also follow the approach of Spred (Liu & Wang, 2023). Specifically, we utilize predetermined thresholds to zero out weights whose absolute values do not exceed these thresholds. Subsequently, we compute the compression ratio of the model alongside its test accuracy on the CIFAR10 test dataset.

Our CIFAR10 experiments can be completed within 3 hours per case on a single 2080Ti GPU.

**Ablation of ReWA components (Figure 3a).** For ReWA, we fix $K = 9$ and ablate the weight decay $\lambda$, adaptive learning rate order $M$, and stabilizer $\varepsilon$, each run 3 times (mean $\pm$ std reported):

- **ReWA w/ M (full, best):** $M = 2$, $\varepsilon = 0$, $\lambda = 10^{-4}$.

- **ReWA w/o M & $\varepsilon$:** $M = 0$, $\varepsilon = 0$, $\lambda = 10^{-4}$.

- **ReWA w/ $\varepsilon$:** $M = 0$, $\varepsilon = 10^{-5}$, $\lambda = 10^{-4}$.

- **ReWA w/o wd, M & $\varepsilon$:** $M = 0$, $\varepsilon = 0$, $\lambda = 0$.

The $\ell_1$ baseline sweeps its regularization coefficient over $\{10^{-5}, 3 \times 10^{-5}, 10^{-4}, 3 \times 10^{-4}, 10^{-3}\}$; the best curve is shown.

### D.3. ImageNet

We adhere to the prescribed training regimen and evaluation framework established by FFCV (Leclerc et al., 2023) for the ImageNet (Deng et al., 2009) dataset. Our training protocol entails training the ResNet50 model for 88 epochs using a batch size of 512, a momentum of 0.9, a label smoothing factor of 0.1, and a cyclic learning rate scheduler. For vanilla SGD, we employ a learning rate of 1.7 and a weight decay of 1e-4. When employing SGD with $\ell_1$ weight decay, we set the $\ell_1$ weight decay to 2e-6 and the learning rate to 1.7. In experiments with $K = 3$, we initialize the learning rate to 1 and set the weight decay to $5e-5$, with $\epsilon$ set to 1e-6.

Our ImageNet experiments can be complete within 4 hours on 8 A100 GPUs for each training task.

**Ablation of ReWA components (Figure 3b).** We fix $K = 3$ and $M = 2$ and ablate weight decay $\lambda$ and stabilizer $\varepsilon$ on ResNet-50/ImageNet, with learning rate 1.0, cyclic schedule, 88 epochs, run 3 times each. The five ReWA variants shown are:

- **ReWA w/ wd**$= 5 \times 10^{-5}$**,** $\varepsilon = 10^{-3}$ (best): $\lambda = 5 \times 10^{-5}, \varepsilon = 10^{-3}$.

- **ReWA w/ wd**$= 5 \times 10^{-5}$**, w/o** $\varepsilon$: $\lambda = 5 \times 10^{-5}, \varepsilon = 0$.

- **ReWA w/ wd**$= 5 \times 10^{-5}$**,** $\varepsilon = 10^{-6}$: $\lambda = 5 \times 10^{-5}, \varepsilon = 10^{-6}$.

- **ReWA w/o wd,** $\varepsilon = 10^{-6}$: $\lambda = 0, \varepsilon = 10^{-6}$.

- **ReWA w/o wd,** $\varepsilon$: $\lambda = 0, \varepsilon = 0$.

Baselines use the same FFCV training pipeline: vanilla SGD (lr= 1.7, wd= $10^{-4}$) and SGD-$\ell_1$ (lr= 1.7, $\ell_1$ coefficient $2 \times 10^{-6}$).

### D.4. CIFAR-10 with AdamW

To verify that ReWA is compatible with Adam-type optimizers, we train ResNet-18 on CIFAR-10 using ReWA combined with AdamW ($K = 9$, $M = 2$, $\varepsilon = 0$, weight decay $10^{-4}$). The accuracy–compression ratio tradeoff is reported in Table 1. ReWA +AdamW maintains 90.30% accuracy at compression ratio $\approx 122\times$, confirming that ReWA is not restricted to SGD-based optimizers.

*Table 1.* ReWA combined with AdamW on CIFAR-10 / ResNet-18 ($K = 9$, $M = 2$, $\varepsilon = 0$, wd= $10^{-4}$).

| Compression Ratio | 10.3× | 52.4× | 82.3× | 122.4× | 157.6× |
|---|---|---|---|---|---|
| Accuracy (%) | 91.61 | 91.58 | 91.45 | 90.30 | 82.33 |

### D.5. Ablation Studies

We consolidate the ablation results here to clearly demonstrate the role of each module in ReWA.

**Role of Reparameterization** ($K$) **and Adaptive Learning Rate** ($M$). $K$ and $M$ jointly govern the sparsity and stability of ReWA, as ReWA induces an $\ell_p$ regularizer with $p = 1 + (1 - M)/K$. Figure 1 illustrates instability when $p$ is small due to the strong gradient of the regularizer. Figures 2b and 2c further confirm this. On CIFAR-10 (Figure 3a), enabling the adaptive learning rate substantially extends the achievable compression ratio.

**Role of Weight Decay** ($\lambda$). Figure 2d shows that larger weight decay $\lambda$ leads to sparser solutions in the linear regime. On CIFAR-10 (Figure 3a), the variant with weight decay ("w/o M & $\varepsilon$", i.e., ReWA with only weight decay) and the variant without any regularization ("w/o wd, M & $\varepsilon$") are comparable. On ImageNet (Figure 3b), comparing "w/o wd" versus "w/ wd=5e-5" variants confirms that weight decay substantially improves the accuracy-sparsity tradeoff.

**Role of Threshold** $\varepsilon$. The experiments verify Proposition 3.4. On CIFAR-10 (Figure 3a), adding $\varepsilon$ ("w/ $\varepsilon$") and the no-regularization baseline are comparable. On ImageNet (Figure 3b), the choice of $\varepsilon$ matters: "w/ wd=5e-5, $\varepsilon$=1e-3" achieves the best compression ratios.

## D.6. Computational Overhead

We report training time and memory usage on the synthetic dataset ($d = 10{,}000$, 800 epochs) in Table 2. The overhead of ReWA is negligible: element-wise $O(d)$ operations are dominated by forward/backward passes.

*Table 2.* Training overhead on the synthetic dataset.

| Metric | SGD-L1 | ReWA |
|---|---|---|
| Total time | 202.1s | 200.1s |
| CPU peak mem | 307.6 MB | 307.8 MB |
| GPU peak alloc | 3.94 MB | 3.98 MB |

## D.7. Hyperparameter Summary and Tuning Guidelines

Table 3 summarizes the best hyperparameter configurations used across all experiments.

*Table 3.* Best hyperparameter configurations for each experimental setting.

| | **Synthetic** | **CIFAR-10** | **ImageNet** |
|---|---|---|---|
| Optimizer | ReWA (SGD) | ReWA (SGD) | ReWA (SGD) |
| $K$ / $M$ / $\varepsilon$ | 9 / 4 / 0 | 9 / 2 / 0 | 3 / 2 / $10^{-3}$ |
| $\lambda$ (wd) | $5 \times 10^{-5}$ | $1 \times 10^{-4}$ | $5 \times 10^{-5}$ |
| Initialization | Kaiming | Kaiming | Kaiming |
| LR / Schedule | $2 \times 10^{-4}$ / Cosine | 0.256 / Cosine (warmup) | 1.7 / Cyclic |
| Epochs / Runs | 800 / 3 | 100 / 3 | 88 / 3 |

**How to select $K$ and $M$ for different models.**

- **Simple models (linear/sparse recovery):** Use Configuration A ($\varepsilon = 0$, $M > 1$), e.g., $K = 9$, $M = 4$.

- **Moderate scale (CIFAR-10/ResNet-18):** $K = 9$, $M = 2$, $\varepsilon = 0$, wd$= 10^{-4}$.

- **Large scale (ImageNet/ResNet-50):** Use Configuration B, $K = 3$, $M = 2$, $\varepsilon = 10^{-3}$, wd$= 5 \times 10^{-5}$.

- **Rules of thumb:** Start with $K \in \{3, 9\}$, $M = 2$; increase $M$ for higher target sparsity; add $\varepsilon \in \{10^{-3}, 10^{-6}\}$ if training is unstable.

