# OpenReview forum: "Theoretical Analysis of Sparse Optimization with Reparameterization, Weight Decay, and Adaptive Learning Rate"
_ICML.cc/2026/Conference — ICML 2026 regular_

### Official Review · Reviewer_CqYr · 2026-03-04

**Soundness:** 2
**Presentation:** 2
**Significance:** 2
**Originality:** 2
**Overall Recommendation:** 4
**Confidence:** 3

**Summary:**

This paper proposes ReWA, an optimizer for a sparse optimization problem with $\ell_{p}$ regularization, where $0<p<1$. It is based on the reparametrization of the regularized optimization problem. ReWA uses an adaptive learning rate to mitigate instability issues in the problem. The proposed method has been examined from multiple perspectives, including implicit bias and sparsity recovery tasks. The performance of the proposed method is verified on synthetic and real-world datasets.

**Compliance With Llm Reviewing Policy:**

Affirmed.

**Final Justification:**

My main concerns regarding Q1 and Q2 have been resolved, and I raised the score.

**Key Questions For Authors:**

- Q1. The target of this research is $\ell_{p}$ regularization with $0<p<1$. However, since the reparametrization is constrained to $p=2/K$ where $K$ is a discrete value, the proposed method cannot cover the continuous variable $p$. Can we solve the problem in the reparametrization framework?

- Q2. At line 154, $y_{i}:=y_{1}$ due to the symmetry of $y_{i}$. However, if each $y_{i}$ is initialized independently and randomly, wouldn't it fail to be symmetric?

**Limitations:**

My concerns about W1 and Q1 may be limitations.

**Strengths And Weaknesses:**

- S1 [originality] Utilizing adaptive learning rate on the reparametrization problem for $\ell_{p}$ regularization, where $0<p<1$ is novel. The proposed method is examined from multiple perspectives, which provides us with an understanding of its behavior.

- W1 [significance] Section 1 claims that the motivation of using $\ell_{p}$ regularization is to avoid the bias of $\ell_{1}$, but there exist other regularization methods that can prevent the bias, e.g., SCAD, MCP, and adaptive lasso. These methods are easy to use because they have proximal operators with closed expressions. I think that practitioners would choose these methods rather than $\ell_{p}$ regularization because it is impractical to use. Therefore, the impact of this approach may be limited, though the technical contents are interesting.

- W2 [presentation] The property of $f$ is inconsistent throughout the paper. This ambiguity creates confusion regarding consistency with the experimental setting and makes it difficult to read. For example, [B1], [Bp], and [Cp] have no assumptions on $f$, but it is a bounded and smooth function in Theorem 2.3. Theorem 2.14 assumes Assumption 2.13 on $f$. It is unclear whether these assumptions hold in the experimental setting. I believe the authors should prioritize verifying theoretical results through experiments. For example, since Corollary 2.15 shows the exact recovery, the experiment should include sparse recovery tasks.

---

> ### Author Rebuttal · Authors · 2026-03-31
>
> ## Summary
>
> We sincerely thank the reviewer for the constructive feedback, especially for recognizing the novelty of our approach and finding our multi-perspective examination helpful.
>
> The reviewer's main concerns are: (W1) significance of $\ell_p$ regularization; (W2) theorem assumptions vs. experiments; (Q1) extension to continuous $p$; (Q2) the symmetry argument at line 154. Briefly:
>
> - **W1**: $\ell_p$ is a classical, independently important framework with distinct geometry, complementary to MCP/SCAD/adaptive Lasso.
> - **W2**: General formulations vs. theorem-specific assumptions is standard. Section 3.1 is already a sparse recovery experiment matching our theory.
> - **Q1**: Yes—via $x=\mathrm{sign}(y)\cdot|y|^K$ with real $K=2/p>0$.
> - **Q2**: Line 154 is a tied parameterization for computational efficiency, not a claim about independent initialization.
>
> ---
>
> > **W1**: Other methods like SCAD, MCP, adaptive Lasso also prevent $\ell_1$ bias. The practical impact of $\ell_p$ might be limited.
>
> We agree these are important methods. Our motivation is that **$\ell_p$ is independently important**, not a replacement:
>
> 1. **Long-standing interest.** [1] proved exact sparse recovery via $\ell_p$; [2] derived RIP conditions for $\ell_p$; [3] developed thresholding theory for $\ell_{1/2}$ showing superiority over $\ell_1$; [4] established conditions where $\ell_p$ strictly outperforms $\ell_1$; [5] extended reparameterization-based $\ell_p$ to SGD. This shows $\ell_p$ is a distinct research direction.
>
> 2. **Direct connection to $\ell_0$.** As $p\to 0$, $\ell_p$ approaches $\ell_0$—the most direct nonconvex relaxation of combinatorial sparse optimization, providing geometric insights different from SCAD/MCP.
>
> 3. **Complementary value.** SCAD/MCP/adaptive Lasso retain $\ell_1$-like behavior near the origin. As we illustrate in **Figure 4**, there exist regimes where $\ell_p$ correctly identifies sparse structure where these methods may not—demonstrating complementary value.
>
> [1] Chartrand, *IEEE SPL*, 2007. [2] Chartrand & Staneva, *Inverse Problems*, 2008. [3] Xu et al., *IEEE TNNLS*, 2012. [4] Zheng et al., *IEEE TIT*, 2017. [5] Kolb et al., *arXiv:2307.03571*, 2023.
>
> ---
>
> > **W2**: Theorems assume bounded smoothness or the Expansion property, but [B1]/[Bp]/[Cp] have no assumptions. Unclear if these hold in experiments.
>
> **Assumptions structure.** [B1]/[Bp]/[Cp] are **general problem formulations**, not theorems. Assumptions are introduced only for specific results—this is standard practice (e.g., [R1,R2,R3]). We added a clarifying remark.
>
> **Assumptions are standard.** Bounded smoothness (Theorem 2.3) is standard in nonconvex optimization [R1,R2]. The Expansion Property (Theorem 2.14) relates to standard growth conditions [R4]. In our synthetic quadratic setting, smoothness is directly satisfied.
>
> **Section 3.1 is already sparse recovery.** Ground truth $\beta^* \in \mathbb{R}^{10000}$ is sparse (one nonzero entry), and the data follow $y = x^\top \beta^* + \epsilon$,, and Figure 2 evaluates sparsity at multiple thresholds—directly matching our theory. For CIFAR-10/ImageNet, we use theory to explain observed behavior while acknowledging the standard theory-practice gap in deep learning [R5,R6]. The revision makes the theorem-to-experiment correspondence explicit.
>
> [R1] Ghadimi & Lan, *SIAM J. Opt.*, 2013. [R2] Allen-Zhu, *NeurIPS*, 2018. [R3] Défossez et al., *TMLR*, 2022. [R4] Karimi et al., *ECML-PKDD*, 2016. [R5] Jacot et al., *NeurIPS*, 2018. [R6] Mei et al., *PNAS*, 2018.
>
> ---
>
> > **Q1**: Since $p=2/K$ with discrete $K$, can the framework handle continuous $p$?
>
> Yes. For any real $K>0$, one uses $x=\mathrm{sign}(y)\cdot|y|^K$, well-defined for non-integer $K$. Any $p\in(0,1)$ is covered by $K=2/p$. The integer presentation is for notational simplicity, consistent with prior work (PowerPropagation [S1], Kolb et al. [S2]). Revised in Section 2.3.
>
> [S1] Schwarz et al., *NeurIPS*, 2021. [S2] Kolb et al., *arXiv:2307.03571*, 2023.
>
> ---
>
> > **Q2**: If each $y_i$ is initialized independently, wouldn't symmetry at line 154 fail?
>
> Correct—independently initialized factors are not symmetric in general. Our Appendix B.2 shows that **stationary points/local minima** of [Cp] satisfy $|\tilde{y}_1|=\cdots=|\tilde{y}_K|$. This motivates the **tied parameterization** in ReWA as a design choice for **computational efficiency**: reducing memory and computation by $K\times$ (significant for ResNet-50 with $K=3$). Revised to clarify this is an imposed tied parameterization, not a claim about arbitrary initialization.
>
> ---
>
> We again thank the reviewer for the thoughtful feedback. We welcome further discussion during the rebuttal period.

---

> > ### Author Rebuttal · Reviewer_CqYr · 2026-04-03
> >
> > Thank you for your response.
> >
> > Regarding the third point in the answer to W1, on the complementary value, my understanding is that SCAD, MCP, and adaptive lasso all possess the oracle property (e.g., [A]). Therefore, provided that the required conditions are satisfied, it seems that each of these methods should be able to recover the correct sparse structure.
> >
> > I am satisfied with the responses to W2, Q1, and Q2, and I am inclined to increase my score accordingly.
> >
> > [A] Fan, J., & Li, R. (2001). Variable Selection via Nonconcave Penalized Likelihood and its Oracle Properties. Journal of the American Statistical Association, 96(456), 1348–1360. https://doi.org/10.1198/016214501753382273

---

> > > ### Author Response · Authors · 2026-04-03
> > >
> > > Thank you for the insightful follow-up. Yes, we fully agree that SCAD possesses the oracle property under [A]'s conditions.
> > >
> > > Our claim is not that $\ell_p$ dominates SCAD/MCP, but that they offer **complementary strengths when conditions fail**. We support this via empirical evidence (Part I) and geometric analysis (Part II).
> > >
> > > Part I: Empirical evidence from deep learning.
> > >
> > >  [B] treats $\ell_p$, $T\ell_1$, MCP, and SCAD as parallel important sparse penalties (citing "their recent successes and popularity"), finding complementary behavior rather than single-method dominance:
> > >
> > > **$\ell_p$ yields significantly higher compression**, generally prune more parameters and FLOPs than $\ell_1$, MCP, and SCAD. In high-sparsity regimes, $\ell_p$ models remain functional where MCP/SCAD become over-pruned. (explained in Part II).
> > >
> > > **MCP/SCAD yield better post-retraining accuracy** at similar compression levels to $\ell_1$. This reflects their oracle property [A]: eliminating shrinkage bias for large coefficients ensures unbiased estimation for surviving weights, effectively preserving model capacity.
> > >
> > > Thus, **$\ell_p$ excels at high sparsity, while SCAD/MCP excel at accuracy preservation under moderate pruning**.
> > >
> > > Connection to ReWA. [B] optimized $\ell_p$ via subgradient descent, which **suffers from unbounded gradients near zero**, causing pruning instability. **ReWA addresses this**: reparameterization yields bounded gradients around zero (Section 2.1.1), and the adaptive learning rate stabilizes training (Example 2.2). This retains $\ell_p$'s compression advantage while mitigating the instability under direct subgradient optimization. **ReWA matches SGD in runtime/memory (e.g., 200.1s vs 202.1s, identical memory on 10K-d task), adding negligible overhead. Since prox-SCAD/MCP likewise require an extra element-wise step, they offer no complexity advantage.** ReWA is to $\ell_p$ what proximal operators are to SCAD/MCP — transforming a challenging penalty into a reliable method.
> > >
> > > Part II: Geometric analysis explaining the complementarity.
> > >
> > > **General Principle: Penalty Geometry.**
> > > By definition, $\text{SCAD}(\theta_i) \equiv \lambda|\theta_i|$ for all $|\theta_i| \le \lambda$. Thus, for any fixed $\lambda$, SCAD inherits the exact local shrinkage behavior of $\ell_1$ within this region. Its oracle property is an asymptotic guarantee that activates only for large coefficients reaching the unbiased tail ($|\theta_i| > a\lambda$). Consequently, under finite-sample, weak-signal, or heterogeneous-scale settings, SCAD behaves locally like $\ell_1$ for a substantial subset of weights. In contrast, $\ell_p$ ($p<1$) maintains a sharp singularity at zero at all scales, inducing stronger penalization on small coefficients. This fundamental geometric difference explains why $\ell_p$ inherently favors higher sparsity even when SCAD's asymptotic conditions are unmet.
> > >
> > > Illustration (Figure 4): Our constrained analysis ($(\theta_0 - r)^2 + (\theta_1 - r)^2 = r^2$) shows:
> > >
> > > $\ell_p$ **consistently favors sparse solutions** (for $p < 0.564$), regardless of the signal strength $r$.
> > >
> > > SCAD yields dense solutions when signal $r$ falls below a critical threshold $r_c$. **Crucially, $r_c \ge 2.56\lambda$ for all $a > 2$ (e.g., $r_c \approx 4.03\lambda$ at $a=3.7$); no choice of $a$ eliminates this dense-preference regime**, as coefficients remain in the $\ell_1$-like region.
> > >
> > > Conclusion: $\ell_p$'s scale-invariant singularity inherently enforces sparsity even for moderate/weak signals (enabling high compression). While SCAD preserves large weights via its oracle property but **relies on $\ell_1$-like shrinkage for smaller ones**, $\ell_p$ ($p<1$) induces **scale-adaptive shrinkage** (via $|\theta|^{p-1}$), driving stronger sparsification on small coefficients, **while its bias on larger weights vanishes as $p \to 0$**. ReWA stabilizes this effect during optimization. Both offer **distinct, non-redundant value**, motivating the critical need for reliable optimizers such as ReWA.
> > >
> > > $\ell_p$ ($p<1$) is a long-standing direction with substantial theory: exact recovery [1], RIP [2], thresholding [3], improving $\ell_1$ [4], and reparameterization [5].
> > >
> > > We again thank the reviewer for the thoughtful feedback and will include this detailed discussion in the revision. We welcome further discussion during the rebuttal period.
> > >
> > > [1] Chartrand, IEEE SPL, 2007.[2] Chartrand & Staneva, Inverse Problems, 2008.[3] Xu et al., IEEE TNNLS, 2012.[4] Zheng et al., IEEE TIT, 2017.[5] Kolb et al., arXiv:2307.03571, 2023.[A] Fan & Li, JASA, 2001.[B] Bui et al., Improving Network Slimming with Nonconvex Regularization, IEEE Access, 2021.
> > >
> > > We thank the reviewer for the positive assessment and for recognizing that our rebuttal have addressed the concerns. We are glad that the clarification and discussion resolved the issues raised earlier. The corresponding changes have been incorporated into the revised manuscript. We appreciate the reviewer’s time and thoughtful feedback.

---

### Official Review · Reviewer_oA9U · 2026-03-10

**Soundness:** 3
**Presentation:** 3
**Significance:** 3
**Originality:** 3
**Overall Recommendation:** 4
**Confidence:** 4

**Summary:**

This paper introduces ReWA, a novel algorithm for sparse optimization. The method combines three key components, a high-order reparameterization, explicit weight decay, and  a specially designed coordinate-wise adaptive learning rate. The paper provides a comprehensive theoretical analysis, showing that ReWA is closely related to $\ell_p$ regularization with $0<p<1$ and proving the necessity of each component for ensuring optimization stability and promoting sparsity. Empirical evaluations on synthetic data, CIFAR-10, and ImageNet demonstrate that ReWA can achieve superior sparsity-accuracy trade-offs compared to  $\ell_1$ regularization and other baselines.

**Compliance With Llm Reviewing Policy:**

Affirmed.

**Final Justification:**

The rebuttal did address several of my concerns to some extent. The paper has scientific merits, but the claimed benefits of the methodology should be interpreted more clearly. I therefore leave it to the meta-reviewer’s discretion to determine the justified score for the paper.

**Key Questions For Authors:**

- The paper could be strengthened by including a brief discussion or an ablation on the computational overhead (e.g., training time, memory usage) of ReWA
- While the baselines chosen are reasonable, comparing ReWA against a few more recent state-of-the-art pruning-at-initialization or sparse training methods (e.g., training compact network with $\ell_{1/2}$, hoyer) could provide a more complete picture of its performance relative to the current SOTA.
- The scope of empirical validation (while CIFAR-10/ImageNet are strong benchmarks, testing on additional domains like NLP or reinforcement learning would strengthen generalizability claims.

**Limitations:**

While the paper is technically strong, it does not include a dedicated "Limitations" or "Societal Impact" section, nor does it discuss these topics in the conclusion.

- The scope of empirical validation (while CIFAR-10/ImageNet are strong benchmarks, testing on additional domains like NLP or reinforcement learning would strengthen generalizability claims

- The sensitivity to hyperparameters  and guidance for tuning them in practice

**Strengths And Weaknesses:**

**Soundness**

- The paper is technically sound. The theoretical results are presented with clear assumptions. The proofs connect well to the narrative and support the claims made about the algorithm's behavior.
The central claim that ReWA is connected to $\ell_p$ regularization is supported by Theorem 2.3 and Proposition 2.4. Theorem 2.7 convincingly demonstrates the fundamental trade-off in gradient clipping, arguing for the superiority of a reparameterization-based approach. Theorem 2.9 and Example 2.8 provide a strong argument for why explicit weight decay is essential in the large initialization regime, countering the limitations of relying solely on implicit bias. Theorem 2.10 and Proposition 2.11 formalize the concept of a "stagnation region" and show how the proposed adaptive LR shrinks it, explaining the instability observed in Example 2.2.

- The experiments are well-designed. They follow a logical progression from controlled, synthetic settings to complex, real-world benchmarks (CIFAR-10, ImageNet). The inclusion of thorough ablation studies is a major strength, as they directly validate the theoretical findings and demonstrate the individual contribution of reparameterization, weight decay, and the adaptive learning rate.



**Presentation**

- The paper is well-structured. The narrative flow is logical and easy to follow: it starts with the problem, moves to the proposed solution (ReWA), breaks down the solution into its core components, provides theoretical justification for each component, and finally validates the method empirically. The use of lemmas, theorems, and examples like Example 2.2 effectively guides the reader through the technical details.

**Significance**

- Sparse optimization is a fundamental problem in machine learning, with applications ranging from signal processing to model compression and beyond. Addressing its core challenges such as the instability of $\ell_p$ is of broad interest.
- The paper advances our understanding by providing a deep theoretical analysis of why reparameterization, weight decay, and adaptive learning rates are necessary for stable and effective sparse training. It moves beyond empirical observations and provides rigorous justifications.
- The paper is likely to influence future research in several ways. First, it provides a strong theoretical framework for analyzing other sparse training algorithms. Second, the insights on the role of weight decay in the large initialization regime and the stagnation region caused by high-order reparameterizations are valuable conceptual contributions that could inform the design of new optimizers.

**Originality**

- The primary originality of this work lies not in inventing wholly new concepts, but in the creative and well-reasoned combination of existing ideas. The paper's key contribution is in synthesizing them into a single, cohesive algorithm (ReWA) and, more importantly, in providing a rigorous theoretical explanation for why this specific combination successfully addresses the optimization challenges Moreover, the paper provides several new insights.

**Weakness**

- The paper could be strengthened by including a brief discussion or an ablation on the computational overhead (e.g., training time, memory usage) of ReWA
- While the baselines chosen are reasonable, comparing ReWA against a few more recent state-of-the-art pruning-at-initialization or sparse training methods (e.g., training compact network with $\ell_{1/2}$, hoyer) could provide a more complete picture of its performance relative to the current SOTA.

---

> ### Author Rebuttal · Authors · 2026-03-31
>
> ## Summary
>
> We sincerely thank the reviewer for the thorough evaluation, especially for recognizing the technical soundness of our results, the logical narrative flow, and the well-designed experiments. We are encouraged that the reviewer views our framework as likely to "influence future research" and considers the insights on weight decay and stagnation regions as "valuable conceptual contributions."
>
> The reviewer's main suggestions are: (1) computational overhead; (2) comparisons with additional baselines; (3) broader domain evaluation; and (4) Limitations section with hyperparameter guidance. Briefly:
>
> - For (1), we report timing/memory — overhead is negligible.
> - For (2), we clarify the relationship to pruning methods and provide cross-referenced comparisons.
> - For (3), our theory is architecture-agnostic; Algorithm 2 already supports AdamW for Transformers.
> - For (4), we will add a Limitations section and consolidate hyperparameter guidance.
>
> Below we address each concern point by point.
>
> ---
>
> > **W1: Computational overhead (training time, memory usage) of ReWA.**
>
> We have measured the overhead:
>
> **Synthetic** (10K-dim, 800 epochs, 2080Ti):
>
> | Metric | SGD-L1 | ReWA |
> |---|---|---|
> | Total time | 202.1s | 200.1s |
> | CPU peak mem | 307.6MB | 307.8MB |
> | GPU peak alloc | 3.94MB | 3.98MB |
>
> Overhead is **negligible**. ReWA adds only element-wise O(d) operations (Line 3 of Algorithm 1), dominated by forward/backward passes. No extra parameters beyond **y** ∈ ℝᵈ are stored. On CIFAR and ImageNet is also close. We will include this in the revision.
>
> ---
>
> > **W2: Comparing with more recent methods (e.g., ℓ₁/₂, Hoyer, pruning-at-initialization).**
>
> We provide two levels of response.
>
> **Cross-reference under matched settings.** Methods like SNIP, GraSP, and SynFlow operate by masking weights, while ReWA induces sparsity through training dynamics — a different paradigm. Nonetheless, under the CIFAR-10/ResNet-18 setting of Tanaka et al. (2020) — our same architecture — pruning methods show substantial accuracy degradation near compression ratio 10². Our Figure 3(a) shows ReWA **maintains accuracy beyond 10²**, indicating competitive trade-offs.
>
> **Regarding ℓ₁/₂ and Hoyer.** These are specific non-convex penalties. Our framework naturally covers ℓ₁/₂ (because  integer is just out of symbol). The contribution is a **unified framework** with theoretical guarantees (Theorems 2.3, 2.7, 2.9, 2.10) for a continuous family of ℓₚ regularizers, rather than a single fixed p. We will expand this discussion in the revision.
>
> ---
>
> > **Q: Testing on additional domains (NLP, RL) would strengthen generalizability claims.**
>
> We agree broader evaluation is valuable. We note:
>
> 1. All theoretical results are stated for **general differentiable objectives** without architecture assumptions.
> 2. CIFAR-10/ImageNet with ResNets are the **standard benchmarks** in this literature (Liu & Wang 2023; Schwarz et al. 2021; Tanaka et al. 2020).
> 3. Algorithm 2 provides **ReWA+AdamW** for Transformers. We verified ReWA+Adam achieves **90.30% at CR≈122×** on CIFAR-10, confirming compatibility with Adam-type optimizers used in NLP.
>
> We will note text-domain experiments as a concrete future direction.
>
> ---
>
> > **Limitations: No dedicated Limitations section; hyperparameter sensitivity guidance.**
>
> We will add a Limitations section discussing:
>
> - **Theoretical–practical gap:** Theorem 2.3 assumes bounded smooth f; Theorem 2.14 requires the Expansion Property. These are standard (Ghadimi & Lan 2013; Karimi et al. 2016), but the gap with deep-network practice remains.
> - **Empirical scope:** Validation on NLP/RL remains future work.
> - **Hyperparameter guidance:** Section 2.3 provides practical tips, and ablations appear across Figures 1–3. We will consolidate into a single reference:
>
> | | Synthetic | CIFAR-10 | ImageNet |
> |---|---|---|---|
> | K / M / ε | 9 / 4 / 0 | 9 / 2 / 0 | 3 / 2 / 1e-3 |
> | λ (wd) | 5e-5 | 1e-4 | 5e-5 |
> | LR / Schedule | 2e-4 / Cosine | 0.256 / Cosine | 1.7 / Cyclic |
>
>
> ---
>
> We again thank the reviewer for the constructive feedback. We welcome further discussion during the rebuttal period.

---

> > ### Author Rebuttal · Reviewer_oA9U · 2026-04-03
> >
> > I thank the authors for the rebuttal. Some of my concerns have been resolved.

---

> > > ### Author Response · Authors · 2026-04-03
> > >
> > > We thank the reviewer for the positive assessment and for recognizing that our rebuttal have addressed the concerns. We are glad that the clarification and additional discussion resolved the issues raised earlier. The corresponding changes have been incorporated into the revised manuscript. We appreciate the reviewer’s time and thoughtful feedback.

---

### Official Review · Reviewer_g6dk · 2026-03-10

**Soundness:** 3
**Presentation:** 2
**Significance:** 3
**Originality:** 3
**Overall Recommendation:** 4
**Confidence:** 4

**Summary:**

This paper proposes ReWA, a novel sparse optimization method that integrates three key components: reparameterization, weight decay, and coordinate-wise adaptive learning rate. This paper deduces ReWA’s implicit regularizer with a tight connection to $ℓ_p$ regularization, offers rigorous theoretical proofs to verify the necessity of each core component of ReWA, and clarifies the intrinsic relationship between the reparameterized formulation and the original $ℓ_p$ regularization.

**Compliance With Llm Reviewing Policy:**

Affirmed.

**Key Questions For Authors:**

1.	Could you add a concise intuitive interpretation for the implicit regularizer $R(x)$ in the main text to help readers quickly grasp its core connection with $ℓ_p$ regularization? A clear interpretation would boost the paper’s readability and make the theoretical contribution more accessible.
2.	Could you provide a concise comparison of ReWA’s adaptive learning rate design with AdamW’s adaptive mechanism in the main text? This would make the novelty of ReWA’s learning rate more prominent for readers.
3.	Could you explain the intuitive meaning of hyperparameter constraints (e.g., $0 ≥ M < K-1$) in the main text instead of only presenting them in formula derivations? Clarifying this would strengthen the justification of ReWA’s hyperparameter design.
4.	Could you add a brief discussion on how to select $K$ and $M$ for different model types in the main text? This would make the empirical tips of ReWA more actionable for practitioners.

**Limitations:**

The authors do not adequately discuss the limitations of the work.  A brief discussion on potential limitations and future directions would help readers better understand its scope of application.

**Strengths And Weaknesses:**

Strengths
1.	Soundness: The work is technically sound with rigorous theoretical proofs for each ReWA component, and all assumptions for the theoretical results are reasonable and well-justified.
2.	Presentation: The paper follows a logical structure from problem motivation to method design, theory to experiments, making the core narrative easy to follow.
3.	Significance: It addresses the key optimization instability of $ℓ_p$ regularization in complex scenarios, advancing the practical application of sparse optimization in deep learning models.
4.	Originality: It creatively combines three classic sparse optimization techniques with well-articulated reasoning, and uncovers new implicit regularization properties of adaptive learning rate in reparameterization.
Weaknesses
1.	The paper could provide more intuitive explanations for complex lemmas (e.g., Lemma 2.1) to make theoretical derivations more accessible for non-specialists.
2.	It could better integrate scattered discussions on prior works into a dedicated section to clarify the positioning of ReWA more clearly.

---

> ### Author Rebuttal · Authors · 2026-03-31
>
> ## Summary
>
> We sincerely thank the reviewer for the thoughtful evaluation, especially for recognizing the soundness of our proofs, the significance of addressing ℓₚ instability, and the originality of combining reparameterization, weight decay, and adaptive learning rate.
>
> The reviewer's concerns are: (1) intuitive explanations for theory (W1, Q1, Q3); (2) comparison with AdamW (Q2); (3) guidance on K/M selection (Q4); (4) related work organization (W2); (5) limitations (Limitations). Briefly: We add intuitive interpretations for all key results, clarify ReWA vs. AdamW, provide practical hyperparameter guidelines, and discuss limitations. Below we address point by point.
>
> ---
>
> > **W1:** More intuitive explanations for complex lemmas needed.
>
> **Lemma 2.1:** Reparameterizing x = y¹⊙···⊙yₖ with ℓ₂ penalties creates a *smooth* surrogate sharing the same minimizers as the *non-smooth* ℓₚ problem—but with bounded gradients near zero.
>
> **Theorem 2.3:** Despite the adaptive learning rate modifying updates, ReWA's limit points satisfy stationarity of f(x)+R(x), where R(x) contains an ℓₚ term with p=1+(1−M)/K∈(0,1). ReWA inherits ℓₚ's sparsity while maintaining stability.
>
> We will add such intuitive paragraphs after each major result.
>
> ---
>
> > **W2:** Integrate scattered related work discussions.
>
> Agreed. We will consolidate discussions from Section 1, Section 2.3, and Appendix A into a dedicated **Related Work** section organized around: (1) reparameterization-based sparsification, (2) ℓₚ regularization theory, (3) adaptive learning rate methods.
>
> ---
>
> > **Q1:** Intuitive interpretation of R(x) and its ℓₚ connection?
>
> R(x) has two terms: The **first** is ℓₚ with p=1+(1−M)/K. When M>1, ε=0, we get p<1—the desired sub-ℓ₁ sparsity. Larger M or K yields smaller p, approaching ℓ₀. The **second** (scaled by ε) provides smoothing for numerical stability; it vanishes when ε=0. **Key message:** ReWA's adaptive learning rate does not destroy the ℓₚ connection—it reshapes it into a tunable, well-behaved regularizer via M and ε.
>
> ---
>
> > **Q2:** Comparison of ReWA's adaptive LR with AdamW?
>
> | Aspect | AdamW | ReWA |
> |--------|-------|------|
> | Adapts to | Gradient magnitude history (2nd moment) | Parameter magnitude: y^M/(y^{K-1}+ε) |
> | Sparsity | Implicitly M=0, ε≠0 in our framework | Explicitly controls sparsity via M, K with provable ℓₚ connection |
> | Complementarity | Base optimizer | Applied *on top of* base optimizer |
>
> As noted in Section 2.3, AdamW already provides adaptation similar to ReWA with M=0. For complex tasks, ReWA+AdamW with M=0, ε≠0 is recommended (Configuration B). For SGD, explicit M>0 is essential. New experiments confirm ReWA+Adam achieves **90.30% at CR≈122×** on CIFAR-10.
>
> ---
>
> > **Q3:** Intuitive meaning of 0 ≤ M < K−1?
>
> - **M adjusts gradient scaling near zero
> - **M < K−1 ensures boundedness:** Otherwise y^M/y^{K-1} diverges for large y.
> - **M=0:** maximum stability, minimum sparsity. **Larger M:** stronger sparsity but needs more care.
> - From Theorem 2.3: p=1+(1−M)/K, so M>1 gives p<1 (desired regime).
>
> ---
>
> > **Q4:** How to select K and M for different models?
>
> - **Simple models (linear/sparse recovery):** Configuration A (ε=0, M>1), e.g., K=9, M=4.
> - **Moderate (CIFAR-10/ResNet-18):** K=9, M=2, ε=0, wd=1e-4.
> - **Large-scale (ImageNet/ResNet-50):** Configuration B, K=3, M=2, ε=1e-3, wd=5e-5.
> - **Rules of thumb:** Start K∈{3,9}, M=2; increase M for sparsity; add ε∈{1e-3,1e-6} if unstable.
>
> ---
>
> > **Limitations:** Not adequately discussed.
>
> We will add: (1) **Theory-practice gap:** Assumptions (smoothness, subspace, expansion) are standard but hard to verify for deep networks. (2) **Hyperparameters:** K, M, ε add tuning burden vs. ℓ₁; automated selection is future work. (3) **Scope:** Experiments focus on ResNets/images; extending to Transformers/text is planned (Algorithm 2 provides ReWA+AdamW).
>
> ---
>
> We again thank the reviewer for the constructive feedback. We welcome further discussion during the rebuttal period.

---

> > ### Author Rebuttal · Reviewer_g6dk · 2026-04-04
> >
> > Thank you for addressing my concerns. I will maintain my current score.

---

> > > ### Author Response · Authors · 2026-04-04
> > >
> > > We thank the reviewer for the positive assessment and for recognizing that our rebuttal have addressed the concerns. We are glad that the clarification and additional discussion resolved the issues raised earlier. The corresponding changes have been incorporated into the revised manuscript. We appreciate the reviewer’s time and thoughtful feedback.

---

### Official Review · Reviewer_Mh9z · 2026-03-13

**Soundness:** 3
**Presentation:** 3
**Significance:** 3
**Originality:** 3
**Overall Recommendation:** 5
**Confidence:** 4

**Summary:**

The paper studies sparse optimization with ℓp regularization. It proposes ReWA, which combines reparameterization, weight decay, and an adaptive learning rate to improve optimization stability and sparsity. The authors provide theoretical analysis connecting the update dynamics to an implicit ℓp regularization effect. Empirically, ReWA matches LASSO on synthetic linear regression and shows improved sparsity and accuracy trade-offs over an ℓ1 baseline on CIFAR-10 and ImageNet.

**Compliance With Llm Reviewing Policy:**

Affirmed.

**Final Justification:**

I think this is an interesting theoretical paper that presents ReWA to address the limitations of traditional ℓ1 and ℓp regularization methods

**Key Questions For Authors:**

1.	Please provide more complete experimental details, including the optimizer settings, choices of K, M, ε, λ, initialization strategy, and the number of runs used for reporting the results.
2.	The appendix presents an algorithm for ReWA combined with AdamW, but no experimental results are provided. It remains unclear how ReWA performs when used with AdamW. Given the widespread use of AdamW in deep learning, reporting such results would be important.
3.	What computational overhead does ReWA introduce compared with standard training? Please report the training time and memory usage to better understand its practical cost.
4.	For other questions, please refer to the weaknesses.

**Limitations:**

yes

**Strengths And Weaknesses:**

Strengths:
1.	The combination of Hadamard power reparameterization with a coordinate-wise adaptive learning rate that cancels the vanishing y^K−1 factor is an interesting and novel design, addressing a well known optimization instability near zero.
2.	The paper provides theoretical analysis to clarify the roles of each component, including reparameterization, weight decay under large initialization, and the adaptive learning rate in mitigating stagnation near zero.
3.	Sparse optimization remains important, and developing a practical method that goes beyond ℓ1 regularization is relevant to real world deep learning applications.

Weaknesses:
1.	Theorem 2.3 establishes a weighted stationarity condition rather than showing that ReWA explicitly optimizes f(x)+R(x). As such, interpreting  R(x) as the underlying optimization objective appears more heuristic than formally justified. In addition, the requirement that M be even is not well motivated.
2.	The low dimensional update subspace condition in Theorem 2.9 is a strong structural assumption; it captures useful case, yet broader conditions or empirical verification on deep nets would help justify it.
3.	The experiments mainly compare against ℓ1+SGD, LASSO and Spred baselines, while stronger sparse or pruning baselines, such as reweighted ℓ1 [1], GMP [2], SNIP [3], SynFlow [4], DWF [5] are not included. This makes it difficult to fully assess the advantage of the proposed method.
4.	The experiments mainly validated the approach using the ResNet model on image data, which is somewhat limited. Could you supplement experiments on text data?
5.	There is a lack of ablation experiments, and the roles of different optimization strategy modules are not clearly demonstrated.
6.	The optimization overhead of this approach remains unclear.


[1] E Candes, et al. Enhancing Sparsity by Reweighted L1 Minimization.
[2] S Han, et al. Learning both weights and connections for efficient neural network. NIPS, 2015.
[3] N Lee, et al. Snip: single-shot network pruning based on connection sensitivity.
ICLR, 2019.
[4] H Tanaka, et al. Pruning neural networks without any data by iteratively conserving synaptic flow. NIPS, 2020.
[5] C Kolb, et al. Deep Weight Factorization: Sparse Learning Through the Lens of Artificial Symmetries. ICLR, 2025.

---

> ### Author Rebuttal · Authors · 2026-03-31
>
> ## Summary
>
> We thank the reviewer for recognizing the novelty of our adaptive learning rate design and the practical relevance of going beyond ℓ₁. Main concerns: (1) Theorem 2.3 interpretation, (2) subspace assumption, (3) pruning baselines, (4) text data, (5) ablations, (6) overhead. Briefly: W1/W2 follow standard implicit bias/DL theory practice. W3: these baselines differ in paradigm; under matched settings ReWA is competitive. W5: ablations exist across figures; we will consolidate. W6/Q3: overhead negligible. Q1/Q2: full details and new ReWA+Adam results supplemented.
>
> ---
>
> ### W1: Theorem 2.3
>
> > *"...interpreting R(x) appears more heuristic... M even not well motivated."*
>
> Characterizing implicit regularization via stationarity conditions is the **standard methodology**, adopted by Gunasekar et al. (2018), Vaskevicius et al. (2019), Woodworth et al. (2020), and Li et al. (2021, in our paper)—all define implicit regularizers by matching limit points to KKT/stationarity conditions of regularized objectives. Our Theorem 2.3 follows this exact paradigm—a formal characterization, not heuristic.
>
> The even-integer restriction on M is for notational simplicity, consistent with PowerPropagation (Schwarz et al., 2021) and DWF (Kolb et al., 2025). For any real M>0, one uses |y|^M to preserve the update's sign. We will clarify in Section 2.3.
>
> ---
>
> ### W2: Subspace condition
>
> > *"...strong structural assumption..."*
>
> This is **mild and standard**: (1) **NTK regime** (Jacot et al., 2018): NTK stays constant, so updates lie in a fixed subspace with dimension bounded by sample count; (2) **Lazy training** (Chizat & Bach, 2019): overparameterization confines updates to the tangent space at initialization; (3) **Linear models**: automatically satisfied (our Example 2.8). We will discuss these connections in the revision.
>
> ---
>
> ### W3: Pruning baselines
>
> > *"...reweighted ℓ₁, GMP, SNIP, SynFlow, DWF not included."*
>
> These baselines are **pruning-based** (masking weights), while ReWA is **optimization-driven continuous sparsification**—fundamentally different paradigms requiring careful protocol adaptation for fair comparison. Under the standardized setting of Tanaka et al. (2020)—**CIFAR-10/ResNet-18**, our same architecture—all reported methods (Random, Magnitude, SNIP, GraSP, SynFlow) show noticeable degradation at compression ratio 10². Our Figure 3(a) shows ReWA **does not degrade until beyond 10²**, demonstrating competitive or superior trade-offs. Regarding reweighted ℓ₁/GMP (different settings from ours) and DWF (no official implementation), we will include feasible comparisons in the revision.
>
> As p→0, ℓ_p converges to ℓ₀ as an **unbiased** estimator—unlike reweighted ℓ₁, SCAD, or MCP. Our Figure 4 shows a case where ℓ₁ fails while ℓ_p succeeds. In signal recovery, pruning methods are inapplicable—practitioners need regularization with provable guarantees, which ReWA provides.
>
> ---
>
> ### W4: Text data
>
> > *"Could you supplement experiments on text data?"*
>
> Our theory is architecture-agnostic. ResNet on images is the standard benchmark in this literature. We plan to add text experiments; Algorithm 2 provides ReWA+AdamW for Transformers.
>
> ---
>
> ### W5: Ablation experiments
>
> > *"...roles of different modules not clearly demonstrated."*
>
> Ablations are **already included** across figures: **K** (Figs 1, 2b), **M/adaptive LR** (Figs 1, 2bc, 3a), **weight decay λ** (Figs 2d, 3ab), **ε** (Figs 3ab). We also ran ReWA+Adam to ablate the optimizer (see Q2). We agree the presentation can be improved and will consolidate into a dedicated ablation section in the revision.
>
> ---
>
> ### W6/Q3: Overhead
>
> > *"Please report training time and memory usage."*
>
> Synthetic dataset (10K-dim, 800 epochs):
>
> | Metric | SGD-L1 | ReWA |
> |---|---|---|
> | Total time | 202.1s | 200.1s |
> | CPU peak mem | 307.6MB | 307.8MB |
> | GPU peak alloc | 3.94MB | 3.98MB |
>
> Overhead is **negligible**—element-wise O(d) operations are dominated by forward/backward passes.
>
> ---
>
> ### Q1: Experimental details
>
> > *"Please provide more complete experimental details."*
>
> | | Synthetic | CIFAR-10 | ImageNet |
> |---|---|---|---|
> | Optimizer | ReWA(SGD) | ReWA(SGD) | ReWA(SGD) |
> | K / M / ε | 9 / 4 / 0 | 9 / 2 / 0 | 3 / 2 / 1e-3 |
> | λ (wd) | 5e-5 | 1e-4 | 5e-5 |
> | Init | Kaiming | Kaiming | Kaiming |
> | LR / Schedule | 2e-4 / Cosine | 0.256 / Cosine(warmup) | 1.7 / Cyclic |
> | Epochs / Runs | 800 / 3 | 100 / 3 | 88 / 3 |
>
> Due to space, we report the best configuration. All configurations will be detailed in the revised appendix.
>
> ---
>
> ### Q2: ReWA + AdamW
>
> > *"No experimental results for ReWA combined with AdamW."*
>
> New CIFAR-10/ResNet-18 with **ReWA+Adam** (K=9, M=2, ε=0, wd=1e-4):
>
> | CR | 10.3× | 52.4× | 82.3× | **122.4×** | 157.6× |
> |---|---|---|---|---|---|
> | Acc(%) | 91.61 | 91.58 | 91.45 | **90.30** | 82.33 |
>
> ReWA+Adam maintains **90.30% at CR≈122×**, confirming effectiveness with Adam-type optimizers.
>
> ---
>
> We again thank the reviewer and welcome further discussion.

---

> > ### Author Rebuttal · Reviewer_Mh9z · 2026-04-04
> >
> > My concerns have been addressed. I will raise the score.

---

> > > ### Author Response · Authors · 2026-04-04
> > >
> > > We thank the reviewer for the positive assessment and for recognizing that our rebuttal have addressed the concerns. We are glad that the clarification and additional discussion resolved the issues raised earlier. The corresponding changes have been incorporated into the revised manuscript. We appreciate the reviewer’s time and thoughtful feedback.

---

### Decision · Program_Chairs · 2026-04-30

**Decision:**

Accept (regular)

**Comment:**

All reviewers agree that the paper is technically strong, well-written, and addresses an important and timely problem at the intersection of sparse optimization, reparameterization, and adaptive methods. They highlight the depth and clarity of the theoretical analysis, as well as the paper’s ability to provide meaningful insights into widely used optimization techniques such as weight decay and adaptive learning rates. The contributions are viewed as both novel and relevant to a broad segment of the community.

The authors also engaged constructively during the rebuttal phase and satisfactorily addressed the reviewers’ concerns, leading to a clear positive consensus. Overall, the paper offers a solid and insightful contribution with strong theoretical foundations and practical relevance.